# Neural signals regulating motor synchronization in the primate deep cerebellar nuclei

Ken-ichi Okada [1,2], Ryuji Takeya[1,2] & Masaki Tanaka [1✉]

Movements synchronized with external rhythms are ubiquitous in our daily lives. Despite the involvement of the cerebellum, the underlying mechanism remains unclear. In monkeys performing synchronized saccades to periodically alternating visual stimuli, we found that neuronal activity in the cerebellar dentate nucleus correlated with the timing of the next saccade and the current temporal error. One-third of the neurons were active regardless of saccade direction and showed greater activity for synchronized than for reactive saccades. During the transition from reactive to predictive saccades in each trial, the activity of these neurons coincided with target onset, representing an internal model of rhythmic structure rather than a specific motor command. The behavioural changes induced by electrical stimulation were explained by activating different groups of neurons at various strengths, suggesting that the lateral cerebellum contains multiple functional modules for the acquisition of internal rhythms, predictive motor control, and error detection during synchronized movements.

[1] Department of Physiology, Hokkaido University School of Medicine, Sapporo 060-8638, Japan. [2] These authors contributed equally: Ken-ichi Okada, Ryuji Takeya. ✉email: masaki@med.hokudai.ac.jp

Synchronized movements to periodic events, such as dancing to music or clapping hands, are common in daily life. Such movements require information processing in the sensorimotor areas in the cerebral cortex, the basal ganglia, and the cerebellum[1–4]. In particular, when the cerebellum is damaged, rhythmic movements cannot be performed with a precision of tens to hundreds of milliseconds[5,6]. This is not surprising since the cerebellum is involved in regulating the timing of contraction of multiple muscles during motor execution, thereby enabling coordinated movements[7,8]. The cerebellum has also been shown to be important in the initiation of movement, especially its timing[9–11]. In theories of motor control, the essential role of the cerebellum is to generate internal models that allow for predictive motor control[12]. The internal models are formed by experience and updated through learning to maintain accurate movement[13]. This adaptive learning mechanism by the cerebellum optimizes not only the magnitude and direction of movement[14–16] but also its timing[17,18].

In addition to motor control, when synchronizing with external rhythms, it is also necessary to detect the periodicity of events and make sensory predictions accordingly. The importance of the basal ganglia in such processes has long been recognized. Evidence from clinical cases[19,20], functional imaging with non-motor rhythmic tasks[21,22], and physiological experiments in primates[23,24] all suggest a role for the basal ganglia in beat-based timing[25]. Recent studies have also shown that the cerebellum is involved in rhythm processing. For example, studies using magnetoencephalography[26] and experimental animals[27,28] have demonstrated that rhythm perception without movement involves periodic entrainment of neuronal activity in the cerebellum and its connected brain regions. Such activities might be related to temporal attention[29,30]. Furthermore, the cerebellum has been shown to be involved in predicting the trajectory of visual stimuli in the absence of movement, indicating that the cerebellum also generates internal forward models for purely sensory events[31–33]. Thus, the cerebellum is likely to be responsible for information processing necessary for synchronized movements, including sensory prediction of periodic events, predictive motor control based on these internal representations, and error monitoring for updating the forward models. Since the cerebellum forms loop circuits with various areas in the cerebral cortex and therefore contains multiple functional modules[34–36], it is possible that these functions are processed in parallel through the relevant networks. However, it remains largely unknown how the cerebellum represents and processes information for synchronized movements.

The present study aimed to investigate this in monkeys performing a series of predictive saccades to targets that alternately appeared on the left and right at regular intervals. Although predictive movements synchronized with external rhythms were previously thought to be behaviours specific to vocal learning species such as humans, songbirds, and dolphins[37], it has recently been shown that monkeys also perform synchronized movements when reinforced with immediate rewards[38,39]. Human functional imaging studies using a similar task have reported increased regional blood flow in the cerebellar crus I during predictive (synchronized) saccades compared to reactive saccades[40]. We examined single neuron activity during synchronized saccades from the posterior portion of the cerebellar dentate nucleus, which receives input from the lateral lobules of the cerebellum and is involved in self-initiated eye movements[41,42]. We found that many neurons exhibited activity correlated with saccade timing and temporal errors, and some responded to eye movements in both directions with enhanced activity during predictive synchronized saccades. The results of electrical stimulation applied to the recording sites indicated that these signals were causally related to the timing of synchronized movements. The posterior portion of the cerebellar dentate nucleus may be part of multiple functional modules involved in the acquisition of internal rhythms, predictive motor control, and error detection.

## Results

**Animal behaviour.** In the synchronized (predictive) saccade task, red saccade targets were alternately presented within horizontally arranged landmarks (white square contours) at a fixed interval (stimulus onset asynchrony or SOA, Fig. 1a). The SOA was randomly selected from 400, 550, and 700 ms for each trial and remained constant during the 12-s trial interval. Since monkeys do not spontaneously generate synchronized saccades to periodic stimuli[43], we rewarded them for every three saccades within ±20% SOA from target presentation to reinforce predictive movements (Fig. 1b, red shadings)[38]. As a control, a reactive saccade task was randomly mixed into the block of trials. In this task, the saccade targets were green, and the duration of each target presentation was randomly selected from 400, 550, and 700 ms. A liquid reward was given for every three reactive saccades (reaction time >100 ms, Fig. 1b, blue shadings).

Fig. 1c plots the temporal error between the stimulus onset and saccade (i.e., saccade latency for reactive saccades) for each condition during the recording experiment in monkey I. In the synchronized saccade task, the temporal error decreased rapidly within a few seconds for all SOAs, indicating that the animal detected rhythmic structure and predictively generated synchronized saccades to the stimulus (Supplementary Movies 1 and 2). In the reactive task, however, the temporal error remained almost unchanged throughout the trial. The latency of the first saccade in the sequence during the recording sessions averaged 305 ± 100 ms (s.d., monkey I) and 291 ± 51 ms (monkey J), while that of the tenth saccade averaged −11 ± 79 ms (monkey I) and −16 ± 87 ms (monkey J) for the synchronized saccade trials and 307 ± 50 ms (monkey I) and 313 ± 73 ms (monkey J) for the reactive saccade trials. Thus, both animals were able to flexibly adjust saccade timing for each task depending on the reward conditions.

**Classification of neurons and recording sites.** During these behavioural tasks, we recorded activity from single neurons responding to saccades in the posterior part of the cerebellar dentate nucleus (Fig. 1d, Supplementary Fig. 1a for all neurons). Quantitative analysis was performed on 95 neurons ($n = 72$ and 23 for monkeys I and J, respectively) for which we were able to examine neuronal activity with a sufficient number (>200) of synchronized saccades. Many of these neurons increased their activity prior to synchronized saccades in one (Supplementary Movie 1) or both directions (Supplementary Movie 2), while the other neurons increased their activity after the saccades. All neurons showed significant changes in activity for saccade direction, the relative time to saccades (± 200 ms), or their interaction (two-way ANOVA, $P < 0.05$). To classify them, a cluster analysis was performed based on the data of each neuron aligned with the synchronized saccades in the 550 ms SOA condition (Fig. 2a, Supplementary Fig. 2a). Approximately 35% ($n = 33/95$) of the neurons exhibited increased activity before saccades in both directions (Bilateral neurons). An equal number of neurons showed clear directional selectivity, with increased activity before saccades ipsilateral ($n = 24$) or contralateral ($n = 9$) to the recording side (Unilateral neurons). The remaining 31% ($n = 29/95$) showed activity mainly after saccades (Post-saccade neurons). When we performed one-way ANOVAs with three factors to compare the firing rate at specific intervals rather than the time course of activity, 89% ($n = 85/95$) of neurons discriminated the direction of saccades and 60% ($n = 57/95$) of

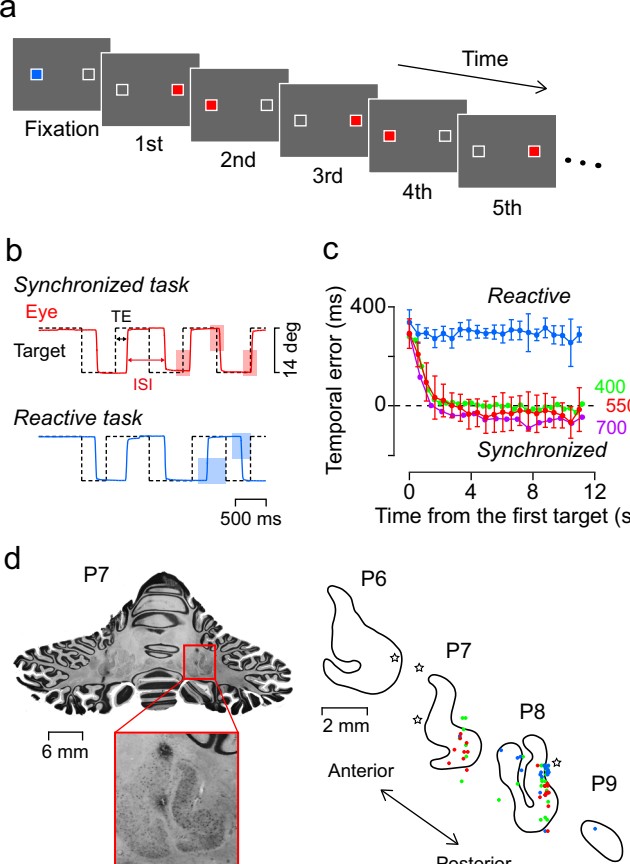

**Fig. 1 Behavioural paradigm and recording sites. a** Saccade targets were presented alternately at landmarks (white squares, 14° apart) that were visible throughout each trial. In the synchronized (predictive) condition, a red saccade target appeared for a fixed interval (stimulus onset asynchrony or SOA) of 400, 550, or 700 ms (constant in each trial), and monkeys were rewarded for predictive saccades (±20% SOA). In the reactive condition, a green target was presented for a random duration, and reactive saccades (reaction time > 100 ms) were reinforced. These conditions were randomly interleaved during each recording session. In both conditions, the stimulus sequence lasted for 12 or 8 s. **b** Sample traces of eye and target positions. Red and blue shadings represent spatiotemporal windows for correct saccades. A liquid reward was given after a random time period (0–600 ms) following three consecutive correct responses. Arrows indicate two temporal parameters of saccades. ISI, inter-saccadic interval; TE, temporal error. The behavioural performance in representative recording sessions is shown in Supplementary Movies 1 and 2. **c** Temporal error (or saccade latency) as a function of stimulus timing in a single recording session. For the reactive condition (blue symbols), data for all target sequences with different SOAs are averaged and plotted every 550 ms. Data from multiple trials are shown as mean ± s.d. ($n = 36$ for the reactive condition, and $n = 36$, 37, and 38 trials for the synchronized 400, 550, and 700 ms SOA conditions, respectively). Numerical data are available in the Source Data file. **d** Histological sections and recording sites in monkey I. Outlines of the dentate nucleus in coronal sections are shown for different posterior locations (P) from the interaural line. Red, green, and blue circles indicate the recording sites of Bilateral, Unilateral, and Postsaccade neurons, respectively. Stars (☆) indicate electrolytic marking lesions. The scale bar in the drawing is common to the enlarged image. A 3-D plot of recording sites in stereotaxic coordinates for all neurons and the depth from the dorsal surface of the cerebellar nucleus is shown in Supplementary Fig. 1.

neurons significantly modulated their activity by the SOA or task condition (Supplementary Fig. 2b).

To characterize the properties of the three types of neurons, we firstly calculated the directional index (DI, Methods) for the data of 550 ms SOA. Unilateral neurons showed strong directional selectivity ($-0.82 ± 0.13$, s.d. and $0.73 ± 0.19$ for neurons with contralateral and ipsilateral preferences, respectively), while Bilateral neurons ($-0.08 ± 0.31$) and Postsaccade neurons ($-0.16 ± 0.46$) showed only weak directional modulation (Supplementary Fig. 3a). The distribution of the DI did not differ between the Bilateral and Postsaccade neurons (Tukey test, $P = 0.78$), while Unilateral neurons showed a different distribution from the others ($P < 0.01$). When the mutual information (MI) was computed to quantify the directional selectivity for each neuron (Supplementary Methods), the MI and DI correlated well ($r = 0.94$), and similar results were obtained (Supplementary Fig. 3d).

We also found that the time course of preparatory activity differed between groups, with the peak timing of Unilateral neurons ($-140 ± 76$ ms, s.d.) being preceded by saccades to a greater degree than that of Bilateral neurons ($-58 ± 87$ ms, unpaired $t$-test, $t_{63} = 4.06$, $P = 0.0001$, Supplementary Fig. 3b). To further assess the time course of preparatory activity, the timing and magnitude of peak activity and the slope of ramping activity were compared across SOAs (Supplementary Fig. 4). Although many individual neurons showed significant changes for different SOAs, the time course of the population activity of Bilateral neurons exhibited a significant change only in the slope of ramping activity and that of Unilateral neurons in the slope and peak timing. When the preparatory activity was temporally scaled, the time course of activity for different SOAs appeared to be very similar for both types of neurons (Supplementary Fig. 4c). The temporal scaling of preparatory activity was consistent with the previous studies in the medial frontal cortex[44–46] and striatum[46,47], as well as those in the cerebellum in the range of hundreds of milliseconds[42].

The recording sites of these groups of neurons differed in the dorsoventral direction (one-way ANOVA, $F_{3,94} = 6.14$, $P = 0.0008$, Supplementary Fig. 1b) but not in the anteroposterior ($F_{3,94} = 1.22$, $P = 0.31$) and mediolateral directions ($F_{3,94} = 1.96$, $P = 0.13$). Postsaccade neurons and Unilateral neurons with a contralateral directional preference were predominantly distributed dorsally in the dentate nucleus, while Unilateral neurons with an ipsilateral preference and Bilateral neurons were widely distributed. This suggests that these neurons may send information to different areas in the brain and are involved in different aspects of behavioural control.

**Enhancement during motor synchronization.** Previous functional imaging studies have shown that cerebellar activity increases during synchronized eye movements[40]. We also found that some neurons in the cerebellar nuclei exhibited differential activity between the synchronized (predictive) and reactive saccades. In the example Bilateral neuron shown in Fig. 2b, the activity was greater during synchronized than reactive saccades. In contrast, in the case of the Unilateral neuron shown in Fig. 2c, the activity before ipsilateral saccades was comparable between the conditions. To compare the activity of individual neurons between conditions, the prediction index (PI) was calculated for each neuron during saccades in the preferred direction (550 ms SOA, Methods). The PI is positive if the firing modulation for synchronized saccades is greater than that for reactive saccades, and negative in the opposite case. The PI averaged $0.10 ± 0.16$ (s.d.), $0.01 ± 0.09$, and $0.03 ± 0.11$ for Bilateral, Unilateral, and Postsaccade neurons, respectively, and only the PIs for Bilateral neurons were significantly greater than zero (one sample $t$-test,

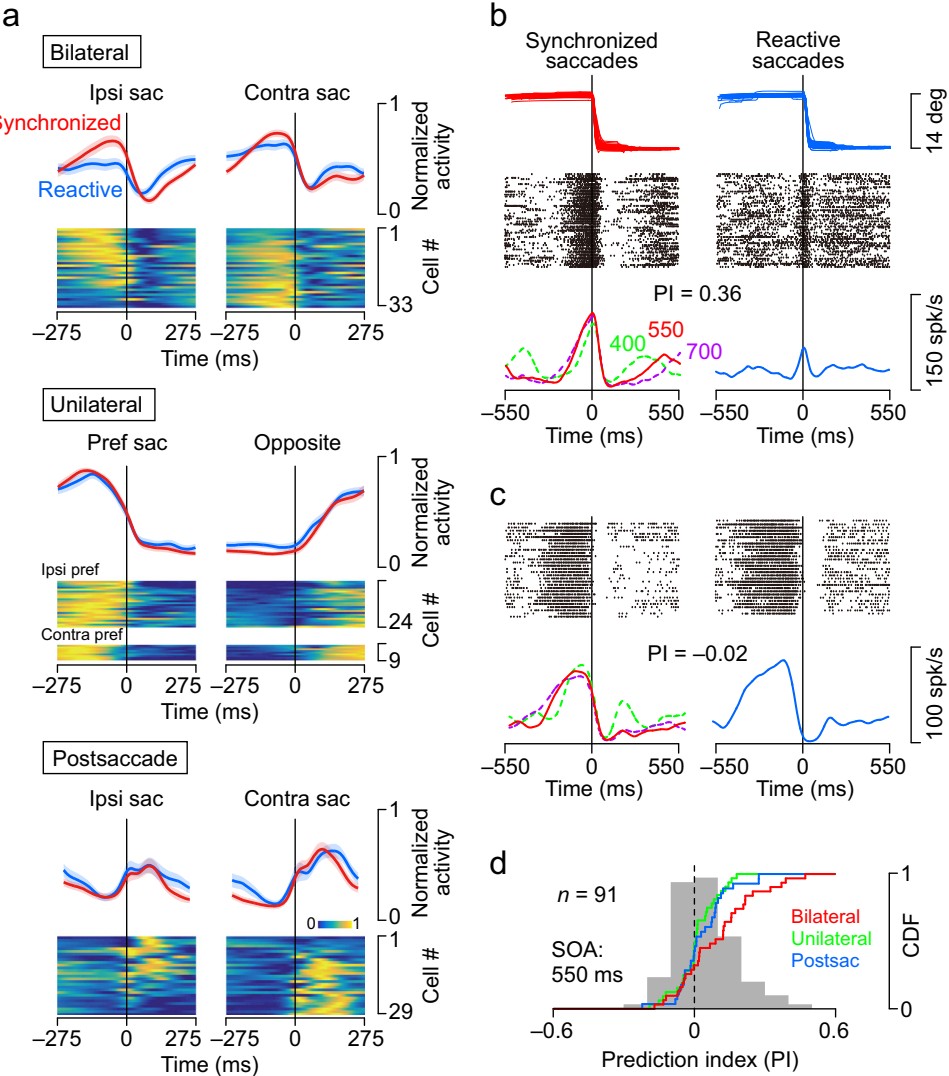

**Fig. 2 Classification of neurons in the dentate nucleus. a** Neurons are classified into four groups according to the time courses of the normalized firing rate aligned with saccades in the opposite directions. Each row illustrates the time course of the population activity (mean ± s.e.) and a heatmap of normalized activity for individual neurons. Yellow and blue indicate higher and lower activity, respectively (*parula* colormap, Matlab). Red and blue traces on each panel indicate the data for synchronized (predictive) saccades and reactive saccades, respectively. Data for neurons exhibiting preparatory activity for either saccade direction were combined (Unilateral type). A clustering dendrogram and the results of ANOVA for different conditions are shown in Supplementary Fig. 2. The directionality of saccade-related responses is summarized in Supplementary Figs. 3a and 5a. Comparison of the time course of preparatory activity between Bilateral and Unilateral neurons across the SOAs are shown in Supplementary Fig. 4. **b** Activity of a Bilateral neuron during synchronized versus reactive saccades. In both panels, the rasters are aligned with ipsilateral saccades for the target with a 550 ms SOA. Green and purple dashed lines on the left panel represent the data for 400 and 700 ms SOAs, respectively. Note the enhancement of activity before synchronized saccades. **c** A Unilateral (contralateral preferred) neuron. As in this example, the peak activity of Unilateral neurons preceded saccades by a greater degree than that of Bilateral neurons (Supplementary Fig. 3b). **d** Comparison of the prediction index (PI) across neuron types. The PI was defined as (*Pred* − *Reac*)/(*Pred* + *Reac*), where *Pred* and *Reac* indicated the firing modulation for predictive and reactive saccades, respectively. The gray histogram represents data from all neurons. The PI was significantly different from zero for Bilateral neurons only (one sample *t*-test, two-sided, $t30 = 3.36$, $P = 0.002$; Unilateral neurons, $t_{31} = 0.64$, $P = 0.53$; Postsaccade neurons, $t_{27} = 1.64$, $P = 0.11$). Similar results were obtained from the mutual information analysis (Supplementary Fig. 3e). The PIs for the other SOAs are shown in Supplementary Fig. 5b. Numeral data are available in the Source Data file.

$t_{30} = 3.36$, $P = 0.002$). We obtained similar results for different SOAs, with the PIs for only Bilateral neurons being greater than zero ($P < 0.02$, Supplementary Fig. 5b). These results suggest that the elevated activity during synchronized saccades found in previous studies might be related to the increased activity of Bilateral neurons (Fig. 2a, top panel). We also found a significant negative correlation between the DI and PI for all neurons ($r = −0.31$, $P = 0.003$, Supplementary Fig. 3c), indicating that neurons with weak directionality tended to have greater activity during synchronized saccades than during reactive saccades.

**Neural correlations with the next saccade timing and the current temporal error**. The lateral cerebellum is essential for the adjustment of saccade timing[42,48]. For both the Unilateral neuron shown in Fig. 3a and the Bilateral neuron in Fig. 3b, the firing rate during a few hundred milliseconds after a saccade varied with the timing of the next saccade. To assess the relationship between neuronal activity and motor timing, we calculated the partial correlation between the trial-by-trial activity of individual neurons and the inter-saccadic interval (ISI), controlling for the previous ISI and temporal error in trials with a 550 ms SOA. The

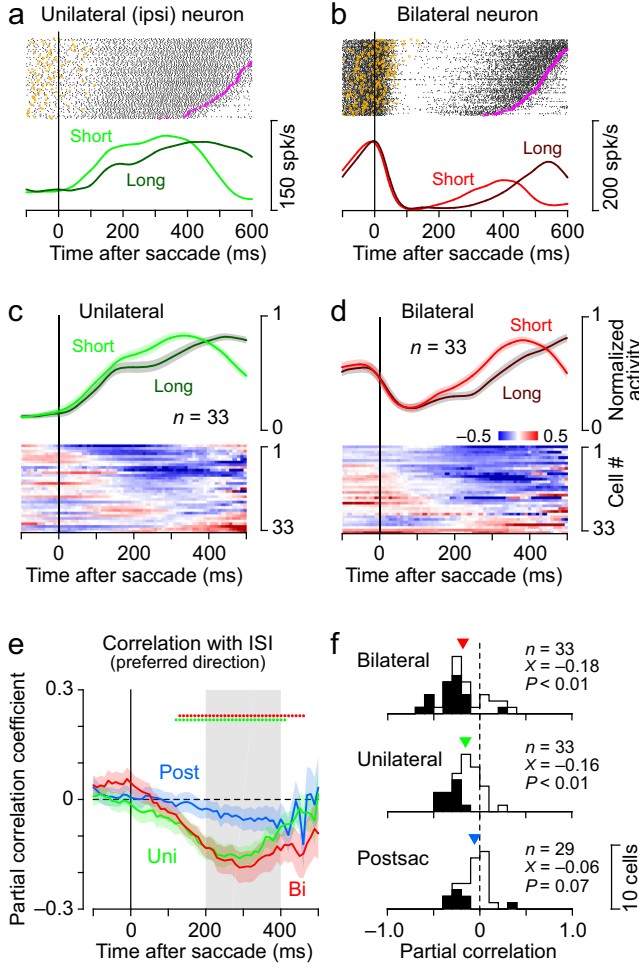

**Fig. 3 Neuronal correlates of the timing of the next saccade. a** A Unilateral neuron with a preference for ipsilateral saccades. The data are from synchronized saccade experiments (550 ms SOA) and trials are aligned with contralateral saccades and sorted by the intersaccadic interval (ISI). Orange and magenta dots on each raster line indicate the onsets of the contralateral target and ipsilateral saccade, respectively. Below the raster plot, the time courses of neuronal activity are separately shown for one-third of trials with longer ISIs and those with shorter ISIs. Note that ramping activity is more advanced in trials with shorter ISIs. **b** A Bilateral neuron (aligned with ipsilateral saccades). **c** Heatmap represents the partial correlation coefficient between the trial-by-trial activity of individual Unilateral neurons and the ISI in trials with a 550 ms SOA. The partial correlation was computed for every 200 ms window (10 ms step) by controlling the temporal error and previous ISI (Supplementary Fig. 6a). The above traces show the mean (±s.e.) of normalized activity in one-third of the trials with short and long ISIs. The same data aligned with the next saccades are shown in Supplementary Fig. 7b. **d** The population activity and partial correlation for individual Bilateral neurons. **e** Time courses of the mean (±s.e.) of the partial correlation coefficients for trials with a 550 ms SOA. The array of coloured dots above the trace indicates when the correlation coefficients significantly differed from zero (one sample *t*-test, two-sided, *P* < 0.05). Gray shading denotes the interval for quantification in **f**. The data for the other SOAs are shown in Supplementary Fig. 5c. **f** Distributions of the correlation coefficients for the Bilateral, Unilateral, and Postsaccade neurons. Black bars indicate data with significant correlations (permutation test, *P* < 0.05). The inverted triangle represents the population mean (X) and the *P*-value on each panel reports the result of the two-sided one sample *t*-test. For this analysis, partial correlation was computed for the preferred saccade direction. Data for contralateral and ipsilateral saccades are shown in Supplementary Fig. 6. Numeral data are available in the Source Data file.

heatmap of the partial correlation coefficients in Fig. 3c indicates that most Unilateral neurons showed a negative correlation at 200–400 ms after the saccade in the non-preferred direction. A comparison of the normalized population activity for one-third of trials with early and late saccades also showed that the elevated activity was associated with a shorter ISI (Fig. 3c, top). The same was true for Bilateral neurons (Fig. 3d), where more than half of them showed a negative correlation. The time course of the partial correlation calculated for the three groups of neurons is shown in Fig. 3e (Supplementary Fig. 5c for the other SOAs). For both Unilateral and Bilateral neurons, the correlation with the next saccade timing appeared at approximately 100 ms after saccades and reached a maximum at ~300 ms, whereas no significant correlation was found in Postsaccade neurons. The partial correlation coefficients 200–400 ms after saccades averaged −0.16 ± 0.16 (s.d.) for Unilateral neurons and −0.18 ± 0.24 for Bilateral neurons, both of which significantly differed from zero (one sample *t*-test, two-sided, Unilateral neurons, $t_{32} = -5.59$ $P = 3.58 \times 10^{-6}$; Bilateral neurons, $t_{32} = -4.38$, $P = 1.20 \times 10^{-4}$; Postsaccade neurons, $t_{28} = -1.89$, $P = 0.07$, Fig. 3f). These correlations between neuronal activity and saccade timing were calculated for saccades in the preferred direction, and similar results were obtained for saccades in the ipsilateral and contralateral directions for both Unilateral and Bilateral neurons (Supplementary Fig. 6). In addition, a significant partial correlation was also observed when the data were aligned with the next saccades (Supplementary Fig. 7b). However, the partial correlation disappeared when the analysis interval was extended backward in time to the previous ISI, indicating that the neuronal activity may regulate the timing of only the next saccades (Supplementary Fig. 7c).

The cerebellum is also thought to be involved in error detection, which is necessary for motor learning. We therefore examined whether neurons in the deep cerebellar nuclei carry information about temporal errors during synchronized saccades. Both the Postsaccade neuron in Fig. 4a and the Bilateral neuron in Fig. 4b showed an increase in post-movement transient activity (inverted triangle) when the saccade preceded the target (yellow dots). To quantify this, we calculated the partial correlation between trial-by-trial neuronal activity and temporal error in individual neurons, controlling for prior ISI in trials with a 550 ms SOA. The results showed that most Postsaccade neurons (Fig. 4c) and about half of Bilateral neurons (Fig. 4d) showed a negative correlation immediately after saccades in the preferred direction. The time course of the correlation coefficients in the three groups of neurons displayed that these negative correlations were maximal at 100–200 ms after saccades and were not observed in Unilateral neurons (Fig. 4e, Supplementary Fig. 5d for the other SOAs). The partial correlation coefficients during 200 ms after saccades averaged −0.10 ± 0.13 (s.d.) for Postsaccade neurons and −0.07 ± 0.27 for Bilateral neurons, and only those for Postsaccade neurons were significantly different from zero (one sample *t*-test, two-sided, $t_{28} = -4.37$, $P = 0.0002$, Fig. 4f). When the same analysis was performed for saccades ipsilateral and contralateral to the recording sites (Supplementary Fig. 8), a significant correlation was found for Postsaccade neurons during ipsilateral saccades ($t_{28} = -4.14$, $P = 2.9 \times 10^{-4}$) and for Bilateral neurons during contralateral saccades ($t_{32} = -2.48$, $P = 0.019$).

We also performed decoding analysis to see if information about saccade timing and task condition can be extracted from each neuron population (Supplementary Methods), and found that all types of neurons carried significant information of the ISI, temporal error and other task conditions at different timing (Supplementary Fig. 9).

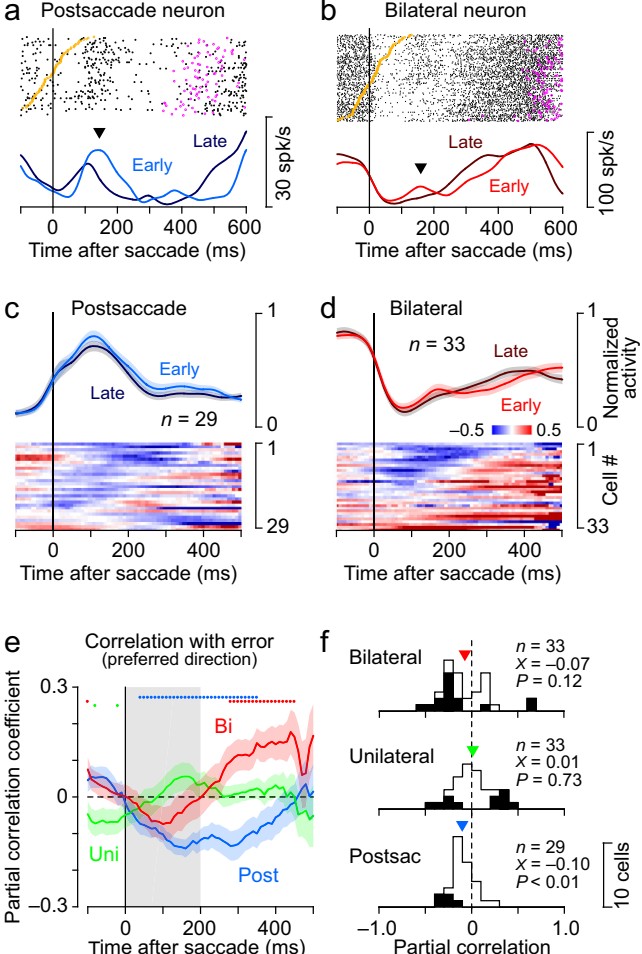

**Fig. 4 Neuronal correlates of temporal error. a** A Postsaccade neuron that responded to the temporal error. The data are from synchronized saccade experiments (550 ms SOA) and trials are aligned with ipsilateral saccades and sorted by temporal error (saccade latency). Orange and magenta dots on each raster line indicate the timing of target onset and the following contralateral saccade, respectively. The time courses of neuronal activity are shown separately for one-third of trials with early and late saccades. Note that the postsaccade activity was large when saccades preceded the target onset (inverted triangle). **b** A Bilateral neuron aligned with ipsilateral saccades. **c** Correlation with temporal error in individual Postsaccade neurons. For each neuron, the partial correlation was computed every 200 ms (10 ms steps) by controlling the preceding ISI (Supplementary Fig. 8a). The above traces indicate the mean (±s.e.) of the normalized population activity. **d** Partial correlation for Bilateral neurons. The traces indicate the mean (±s.e.) of the normalized population activity. **e** Time courses of the partial correlation coefficient (mean ± s.e.) for different types of neurons. The array of coloured dots above the trace indicates when the correlation coefficients significantly differed from zero (one sample t-test, two-sided, $P < 0.05$). The data for the other SOAs are shown in Supplementary Fig. 5d. **f** Distributions of the correlation coefficient for individual neurons. Partial correlation was measured during the interval denoted by the gray shading in **e**. The black bar indicates significant correlation (permutation test, $P < 0.05$). The inverted triangle represents the population mean (X) and the P-value indicates the result of the two-sided one sample t-test (Bilateral neurons, $t_{32} = -1.60$, $P = 0.12$; Unilateral neurons, $t_{32} = 0.34$, $P = 0.73$; Postsaccade neurons, $t_{28} = -4.37$, $P = 1.5 \times 10^{-4}$). Data for all types of neurons sorted by either saccade direction are shown in Supplementary Fig. 8. The results of decoding analysis for the ISI, temporal error, and the other task conditions are shown in Supplementary Fig. 9. Numeral data are available in the Source Data file.

**Separation of sensory and motor responses.** Recent studies have shown that the cerebellum is also involved in sensory prediction without movement execution[27,31]. Is the activity of the different groups of neurons associated with periodic sensory prediction necessary for synchronized saccades, or solely associated with motor execution? Since synchronized saccades are made at the predicted timing of target presentation, it is difficult to separate them. To address this, we focused on neuronal activity during the transition from reactive to synchronized saccades at the beginning of each trial (Fig. 1c), where sensory prediction may precede saccade execution.

The Unilateral neuron shown in Fig. 5a displayed strong activity before leftward saccades, with the peak of the activity gradually preceding the target as the saccade reaction time shortened for the first, third, and fifth leftward targets (vertical red lines). In the right panel, the activity aligned with the first, second, third, fourth, and subsequent targets and saccades show that this neuron consistently exhibited increased activity before leftward saccades (the second column with orange shading). In contrast, the Bilateral neuron in Fig. 5b exhibited a transient decrease in activity after the first target onset, followed by an increase in activity that peaked at the time of the second target (the first vertical blue line). Subsequently, the activity peaked around the time of the target presentation with some directional selectivity. When the data were aligned with either target onset or saccades (right panel), the neuronal activity was more consistent with the target appearance than with saccade initiation.

To assess whether the activity of each neuron was better aligned with sensory prediction or saccade, we calculated the sensorimotor index (SMI, Methods) for the second saccades. The SMI was negative if the neuronal activity was associated with saccades and positive if it was associated with the visual stimulus, and the index was calculated to be −0.90 and 0.27 for the neuron shown in Fig. 5a and b, respectively. As shown in Fig. 5c, the SMIs of most Unilateral and Postsaccade neurons were negative, with means (s.d.) of −0.45 ± 0.34 and −0.33 ± 0.40, respectively. In Bilateral neurons, about half showed a positive SMI, with a mean of −0.04 ± 0.41, which was significantly greater than that of the other types of neurons (Tukey test, $P < 0.02$). The SMI calculated for saccades in the non-preferred direction also differed between Postsaccade and Bilateral neurons (unpaired t-test, $t_{46} = 3.52$, $P = 0.001$, Supplementary Fig. 10b). These results suggest that Bilateral neurons may play a role in sensory prediction whereas Unilateral and Postsaccade neurons are involved in motor control.

To further confirm these findings, we performed three additional analyses. First, the similarity of the time course (rather than the magnitude) of neuronal activity was examined by calculating a correlation coefficient between the data for the second reactive saccades and later synchronized saccades (>5th). When the data were aligned with saccades, Unilateral neurons showed more consistent response than Bilateral neurons (Supplementary Fig. 10c, left panel). However, when the data were aligned with the target onset, Bilateral neurons exhibited more consistent response than Unilateral neurons (right panel).

Second, we applied the time-warping analysis that was previously used to dissociate sensory from motor responses during synchronized tapping[49] (Supplementary Methods). Briefly, this analysis quantifies the likelihood of spike occurrence under the assumption of sensory or motor alignments. Although all neurons showed a preference for motor alignment during synchronized saccades (because movement occurred at the predicted rather than actual sensory timing), only Bilateral neurons significantly changed their preference toward sensory alignment during the second saccades (paired t-test, two-sided, $t_{32} = 3.19$, $P = 0.003$, Supplementary Fig. 11a).

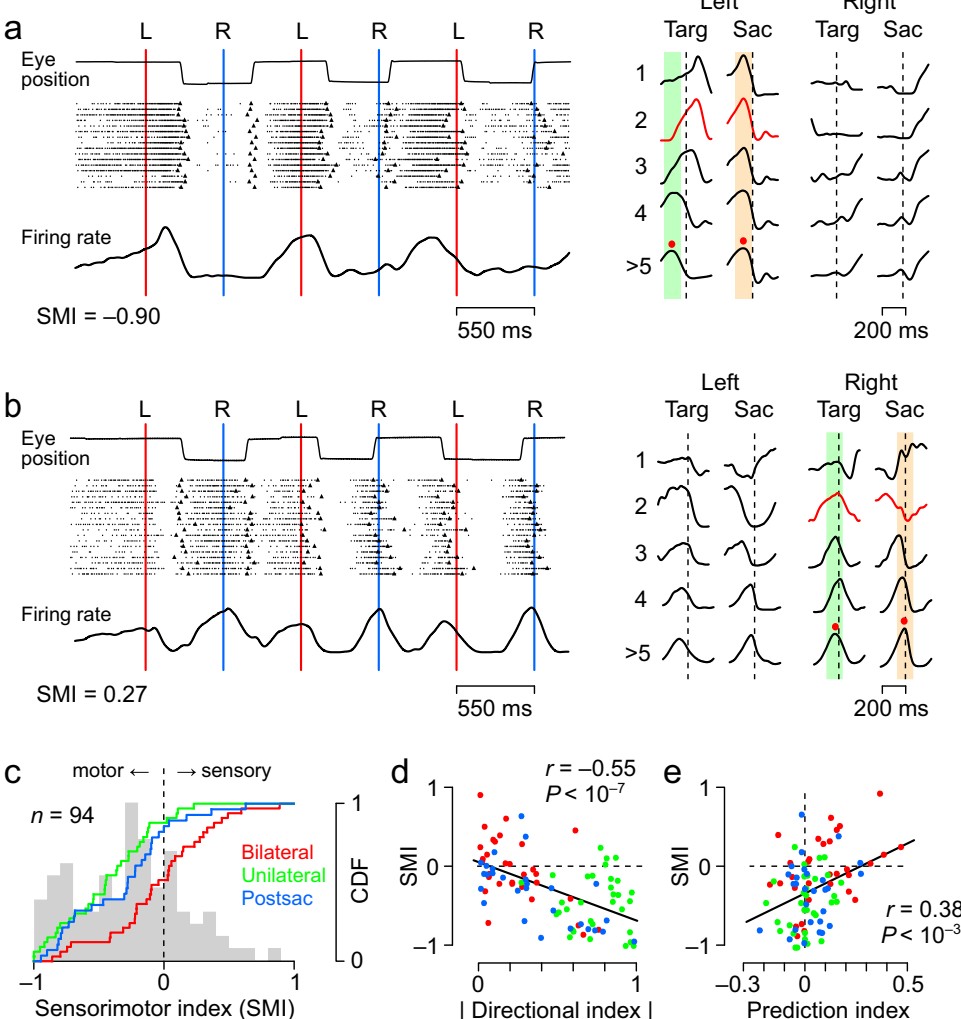

**Fig. 5 Dissociation between sensory prediction and motor preparation. a** *Left*: Response of a Unilateral neuron to the early stimulus sequence. Data are aligned with the target onset (blue and red vertical lines) in the synchronized (predictive) saccade trials with a 550 ms SOA. The black symbol on each raster line indicates saccade onset. Note that neuronal activity consistently preceded the leftward saccade rather than the target onset. *Right*: Traces of the mean firing rate aligned either to the target onset or saccade initiation. Numbers on the left denote the stimulus sequence. Green and orange shadings represent the 100 ms window centred at the peak activity during the later stimulus sequence (red dot on the bottom trace). Red curves indicate the responses to the second target or saccade in the preferred direction, which were used for the quantitative analysis in **c**. **b** Response of a Bilateral neuron. Note that the peak of the neuronal firing rate roughly aligned with target onset even during the initial few cycles of target presentation, while saccades lagged behind the target onset. **c** Comparison of the sensorimotor index (SMI) across neuron types. The SMI was computed for the response to the second stimulus in the sequence as $(Targ − Sac)/(Targ + Sac)$, where *Targ* and *Sac* indicated the activity for target onset and saccades, respectively. The value becomes positive as neuronal activity is better aligned to the target onset than to saccade initiation. Since the SMI became close to zero as the animals generated synchronized saccades (Supplementary Fig. 10a), we focused on the SMI for the second saccades. The SMI computed for saccades in the non-preferred direction is shown in Supplementary Fig. 10b. **d, e** Relationship between the indices. The colour of the dots indicates the type of neuron. The relationship between the three indices is shown in Supplementary Fig. 11c. Numeral data are available in the Source Data file.

Finally, we also examined the change in the time course of target-aligned neuronal activity with the order of target presentation (Supplementary Fig. 11b). When the difference in the time course of activity for Unilateral neurons between the first and the later cycles was quantified using the ROC analysis (Supplementary Methods), the area under the curve (AUC) averaged $0.71 ± 0.15$ (s.d.) and was significantly different from 0.5 (one sample *t*-test, two-sided, $t_{32} = 8.26$, $P < 10^{-8}$). In contrast, for Bilateral neurons, the activity lasted longer in the first cycle, but the centre of the activity was almost the same as that during synchronized saccades (AUC, $0.51 ± 0.15$, $P = 0.66$). These results further suggest that Unilateral neurons are involved in saccade generation, while Bilateral neurons may reflect the temporal prediction

of target appearance both in the initial cycles and during synchronization.

When the relationship between the SMI and the absolute value of the DI was examined for all neurons, there was a significant negative correlation between them ($r = −0.55$, $P = 1.5 × 10^{-8}$, Fig. 5d). There was also a significant correlation between the SMI and PI, with neurons that increased their activity during synchronized movements reflecting the sensory component more strongly ($r = 0.38$, $P = 1.8 × 10^{-4}$, Fig. 5e). The relationships between these three indices are summarized in Supplementary Fig. 11c.

**Causal role in synchronized saccades**. To determine the causal relationship between these neuronal activities and behavioural

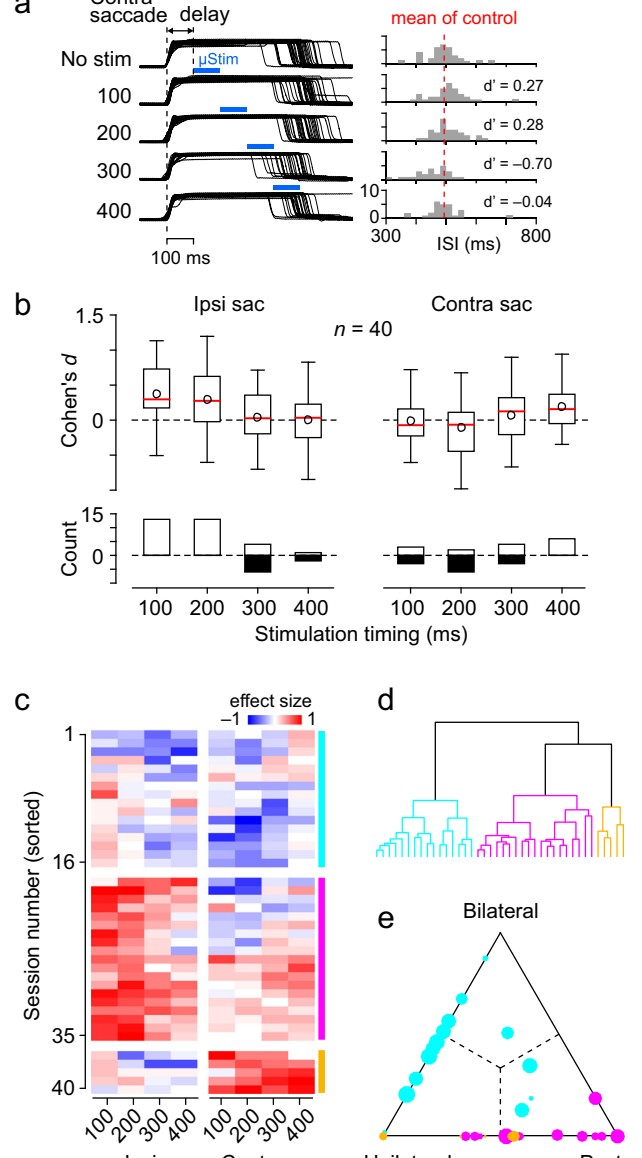

**Fig. 6 Effects of electrical microstimulation. a** *Left*: Traces of eye position in a representative session. A train of biphasic pulses of stimulation current (100 μA, 333 Hz for 100 ms) was delivered through an electrode at different times following the fifth or later saccade during the synchronized saccade trial. The numbers indicate the stimulation timing in milliseconds. *Right*: Histogram shows the distribution of the ISI in each condition. Red vertical dashed line represents the mean for the non-stimulation control. d′ indicates the effect size (Cohen's *d*) for each condition. **b** The box-whisker plot in the upper panel summarizes the median, quartiles, and range of the effect size on the next saccade timing for different stimulation times and saccade direction. Circles indicate the respective mean value. The lower panel summarizes the number of sessions with a significant stimulation effect (Dunnett test, *P* < 0.05), separately showing the facilitatory (negative values) and suppressing (positive) effects. The stimulation effects on the timing of two later saccade are shown in Supplementary Fig. 12. The stimulation effects are separately plotted for dorsal and ventral sites in Supplementary Fig. 13f. **c** Effects of electrical stimulation in individual sessions. For each site, the stimulation effects on the ISI are shown for different stimulation times and saccade direction. Sessions are sorted based on the clustering analysis, and the colour bars on the right indicate different groups. Depths of stimulation sites for each cluster are shown in Supplementary Fig. 13e. **d** A dendrogram derived from the cluster analysis. **e** Relative contribution of the different types of neurons to the stimulation effect. For each type of neuron, the impact of stimulation was estimated by computing the simple correlation coefficients between the neuronal activity and the ISI at different stimulation times (100 ms interval, Supplementary Fig. 13a–c). The contributions of different types of neurons were calculated by fitting the data of stimulation effects in **c** with the impact of neuronal activity on saccade timing shown in Supplementary Fig. 13c. Colour and size of circles indicate the cluster number and the goodness-of-fit summarized in Supplementary Fig. 13d, respectively. Numeral data are available in the Source Data file.

performance, electrical microstimulation was applied to the recording sites during synchronized saccades (550 ms SOA). A train of stimulation pulses (100 ms in duration at 333 Hz, 100 μA) was delivered 100, 200, 300 or 400 ms after the saccade, and the effects on the next saccade timing (or ISI) were examined (Fig. 6a). The box-whisker plot in Fig. 6b summarizes the effect sizes for all stimulation conditions for the 40 sites that showed significant effects in any of the conditions (Dunnett test, *P* < 0.05, *n* = 32 and 8 for monkeys I and J, respectively). The stimulation affected only the timing of the next saccades, and monkeys adjusted after the next ISI to compensate temporal errors induced by the stimulation (Supplementary Fig. 12).

The effects of electrical stimulation varied greatly from site to site, and significant facilitatory and suppressive effects on saccades were found in various proportions, even under the same conditions (Fig. 6b). To examine the differences by stimulation site in detail, the data were clustered into three categories based on the effect size in each condition (Fig. 6c and d). In the first group, electrical stimulation promoted saccades in both directions (vertical cyan bar on the right in Fig. 6c). In the second group, stimulation facilitated contralateral but delayed

ipsilateral saccades, and the effect was greater when stimulation was delivered immediately after the previous saccade (magenta). In the third group, only contralateral saccades were delayed regardless of stimulation timing (orange).

We assumed that the changes in saccade timing for each site could be explained by a certain combination of the effects of stimulation to the three types of neurons (Methods). We estimated the relative impact of electrical stimulation to each group of neurons by calculating simple Pearson correlations between neuronal activity at specific 100 ms periods and the ISI (Supplementary Fig. 13a–c). Our model well explained the stimulation effects such that the coefficients of determination (CDs) were significantly greater than those obtained from the permutation data (1000 iterations, Wilcoxon's rank-sum test, *P* < 0.01, Supplementary Fig. 13d) and the CDs for 72.5% of the stimulation sites were greater than the median of the permutation data. The triangular plot in Fig. 6e summarizes the relative contribution of each neuronal population to the changes in saccade timing derived from the fitted data. The data showed that the first group (cyan) was mainly accounted for by the stimulation effects of Bilateral and Unilateral neurons, and that the second group (magenta) was due to the Postsaccade neurons. In addition, Bilateral neurons and Unilateral neurons were stimulated in various relative proportions, while Bilateral neurons and Post-saccade neurons were rarely stimulated simultaneously. This observation was consistent with the results of our neuronal recordings, which showed that Bilateral neurons and Postsaccade neurons were distributed separately, mainly in the ventral and dorsal portions of the dentate nucleus, respectively (Supplementary Fig. 1). In fact, the stimulation sites for the second group tended to be distributed more dorsally than the other groups (one-way

ANOVA, $F_{2,39} = 3.37$, $P = 0.045$, Supplementary Fig. 13e). Consistent with these findings, the effects of stimulation at dorsal and ventral sites showed similar properties to the second and first groups, respectively (Supplementary Fig. 13f).

Although we did not deliver electrical stimulation during reactive saccades, the sizeable neuronal activity between conditions in Unilateral and Postsaccade neurons (Fig. 2d) suggests that electrical stimulation of these neurons may change the latency of reactive saccades. Indeed, the activity of these neurons before target presentation in the reactive task correlated with saccade latency in a similar way to those during synchronized saccades (Supplementary Fig. 14). The correlation analysis also showed that the activity of Bilateral neurons inversely correlated with the latency of contralateral reactive saccades but not with the latency of ipsilateral saccades, whereas the same neurons were significantly correlated with the timing of synchronized saccades in both directions (Supplementary Fig. 13b). These results indicate that the signals conveyed by Bilateral neurons may not be strictly related to movement like the other types of neurons. Furthermore, the correlation with temporal error after saccades, normally found in Postsaccade neurons during synchronization (Fig. 4e), disappeared during reactive saccades. These results suggest that electrical stimulation to the ventral part of the dentate nucleus (where Bilateral neurons reside) may promote only contralateral reactive saccades, while the same stimulation promoted synchronized saccades in both directions (Fig. 6c and Supplementary Fig. 13f).

## Discussion

Synchronized movement requires multiple neural processes, including adjustment of movement timing, detection of temporal error, and generation of internal rhythms. We have shown that all these signals for synchronized saccades are represented in the posterior part of the cerebellar dentate nucleus. Unilateral neurons mostly showed ipsilateral directional preference and exhibited preparatory activity that closely correlated with the timing of the next saccade. Postsaccade neurons showed transient activity immediately after saccades, and about half of them correlated with temporal error of synchronized saccades. Bilateral neurons, which made up about one-third of the recorded neurons, had both of these properties. Of the three types of neurons, only Bilateral neurons displayed greater activity during synchronized saccades than during reactive saccades, which might be related to the increased activity reported in the previous imaging study[40].

During synchronization, the brain needs to maintain an internal rhythm to make a series of predictive movements[50]. Since our monkeys had been trained on the synchronized saccade task for many months, it is likely that they were able to quickly generate an internal rhythm in each trial. In fact, we have previously shown that the animals can continue to make saccades for many cycles even after removing the temporal error by presenting a target at the time of eye movements (error-clamp condition)[38]. We found in this study that Unilateral and Postsaccade neurons consistently increased their activity in relation to saccades, but many Bilateral neurons were active at the time of target onset during the transition from reactive to synchronized saccades in each trial (Fig. 5). These results suggest that Bilateral neurons may represent the expected timing of target onset, rather than promoting a specific motor output.

Previous studies have shown that internal rhythms emerge in predicting the appearance of periodic stimuli even in the absence of movement[51], and that the cerebellum is involved in this process[26]. In particular, recent studies in primates have demonstrated that neurons in the posterior portion of the dentate nucleus exhibit predictive activity that peaks at the timing of

periodic visual stimuli[27,52]. Furthermore, evidence shows that the cerebellum also plays a role in generating internal forward models for purely sensory aperiodic events, such as trajectory of slowly moving objects[31–33,53]. The activity of Bilateral neurons described so far could be regarded as an internal model of periodically alternating visual stimulus, which guides synchronized saccades.

The three types of neurons were localized in a restricted region of the cerebellar dentate nucleus. While there was no difference in the anterior-posterior or medial-lateral coordinates of the recording sites between the groups, there was a clear difference in the depth of the recording sites (Supplementary Fig. 1). Since the stimulation effects on saccade timing can be explained as a combination of the effects of excitation of these groups of neurons, each group may be involved in multiple processes necessary for synchronized saccades in parallel, by sending information to different brain regions. Neurons in the posterior portion of the cerebellar dentate nucleus are known to project directly to the superior colliculus (SC)[54] and indirectly via the thalamus to the frontal eye fields (FEF), supplementary eye field (SEF), and posterior parietal cortex (PPC)[34]. Recent studies using trans-synaptic tracers have reported that the FEF and PPC receive signals mainly from the ventral portion of the dentate nucleus[34,55], whereas the trans-thalamic projections to the SEF originate from both the dorsal and ventral portions[56].

Supplementary Fig. 15 shows a diagram of the hypothesis that the cerebellum controls synchronized saccades through multiple loops with oculomotor areas in the cerebral cortex. Bilateral neurons participate in the link with the SEF and represent the temporal prediction of the target sequence. Postsaccade neurons participate in the link with the PPC and SEF, which plays a role in monitoring errors and updating internal rhythms. Unilateral neurons participate in the link with the FEF and/or SC and regulate saccade timing. Although this multiple-loop hypothesis is highly speculative, it will provide a practical working hypothesis for future studies exploring signals in these cortical areas and the SC during synchronized saccades.

Two issues must be considered when interpreting the present results. First, the possible involvement of the medial part of the cerebellum in synchronized saccades cannot be ruled out. For oculomotor control, vermal lobules VI–VII regulate signals in the brainstem saccade generator through the fastigial nucleus[57]. This pathway is important for the adaptation of saccade amplitude and direction[15], but its involvement in saccade timing might become apparent in future studies. For example, recent studies in rodents have shown that both the medial[58] and lateral[59] cerebellum are involved in self-timed somatic movements. The second issue is that in the present study, the animals alternated saccades in opposite directions, whereas many studies of synchronized movement employ tasks with a repetition of identical movements, such as tapping. However, everyday activities such as dancing and clapping involve adjusting the timing of multiple movements, and many similarities can be found with synchronized eye movements. The present results may be generalized by exploring neuronal activity in the medial frontal cortex and the striatum, which has been examined during tapping[24,60], using the synchronized saccade task.

## Methods

**Animal preparation**. Two adult male Japanese macaques (monkeys I and J, 7–9 years old, 8–9 kg) were used. These animals were previously used in a series of behavioural experiments[38,61,62]. All experimental protocols were evaluated and approved in advance by the Hokkaido University Animal Care and Use Committee and were in accordance with the Guidelines for Proper Conduct of Animal Experiments (Science Council of Japan, 2006). Animal health and well-being were carefully monitored by animal care staff and experimenters, and food intake, water supply, stool volume, and overall physical condition were checked and recorded

daily. To motivate the animals to perform the tasks, their water intake was regulated during weekday training and experiments, but they had free access to water on weekends. There was no strict dietary restriction, and a variety of vegetables, fruits, nuts and grains were provided daily.

The procedures for animal surgery and recording experiments were identical to those described previously[63]. Briefly, in separate surgeries under general isoflurane and nitrous oxide anaesthesia, a pair of plastic head holders was installed to the skull using titanium screws and dental acrylic, and a scleral search coil was implanted under the conjunctiva. Analgesics were administered during each surgery and for the following few days. After recovery from surgery, the monkeys were trained in the synchronized saccade task[38] for several months. During the training and the subsequent experimental sessions, the monkey's head was secured to the primate chair in a darkened booth, and horizontal and vertical eye position were recorded using the search coil technique (MEL-25, Enzanshi Kogyo). After training on eye movement tasks, a third surgery was performed to place a recording chamber for vertical electrode penetration aimed at the deep cerebellar nuclei. The location of the chamber was verified postsurgically using MRI. Daily recording sessions began after full recovery from the surgery. Topical or systemic antibiotics were administered as necessary.

**Visual stimuli and behavioural task**. Experiments were controlled by a Windows-based stimulus presentation and data acquisition system (TEMPO, Reflective Computing). Visual stimuli were presented on a 27-inch liquid crystal display monitor (XL2720Z, BenQ, refresh rate 120 or 144 Hz) that was located 40 cm away from the eyes and subtended $73° \times 46°$ of visual angle. Throughout the experiment, two landmarks (white unfilled 1° squares) were presented ±7° horizontally (Fig. 1a) on a dark background, and all visual stimuli were presented within the landmark. Each trial started with the appearance of an initial fixation point (blue or purple square, 10.9 cd/m$^2$) at either landmark location. After a random 1000–2000 ms period, the fixation point was extinguished and a saccade target (red or green square, 33.9 cd/m$^2$) was presented at the other landmark location. The saccade target was alternately presented at the landmark locations with a stimulus onset asynchrony (SOA; 400, 550, or 700 ms) for 8000 or 12000 ms (11–30 target steps) and was visible until the appearance of the opposite target. The animals were trained to follow the alternating targets with their eyes. The trial was aborted immediately if the monkey made an anticipatory saccade to the first target in the sequence (reaction time < 100 ms), or if the inter-saccadic interval (ISI) was shorter than 25% SOA or longer than twice of the SOA, or if the eyes were deviated >3.5° vertically from the target locations.

The animals performed the task under two different stimulus conditions. In the synchronized (predictive) condition (Fig. 1b, top), the initial fixation point was blue, the saccade target was red, and the SOA was constant in each trial but varied from trial to trial. To promote synchronized saccades, a liquid reward was given after every three consecutive predictive saccades (generated within ±20% SOA of the target appearance) for the fourth or subsequent targets in the sequence. The reward was delivered at a random time within 600 ms after every three synchronized saccades. Because the initial fixation period and SOA varied from trial to trial, the monkeys were unable to predict the timing of the first two stimuli. In the reactive condition (Fig. 1b, bottom), the initial fixation point and saccade target were purple and green, respectively, and each SOA was randomly selected from 400, 550, and 700 ms within each trial. Animals received an immediate reward after every three reactive saccades that were generated between 100 ms after the target onset and the appearance of the opposite target. In both conditions, the amount of a single reward was adjusted so that the total reward for each trial was approximately the same across the SOAs. These two conditions were presented randomly in a block of trials. During the recording sessions, reactive saccade trials were presented with equal probability to synchronized saccade trials of each SOA, although four neurons (4.2%) were tested for synchronized saccades only. Because both monkeys had previously been trained on a similar synchronized saccade task[38,62], they quickly learned to switch behavioural strategies on each trial within a few days.

**Recording procedures**. Single neuron activity was recorded from three dentate nuclei of the two monkeys. Single tungsten microelectrodes (~1.0 MΩ at 1 kHz, Alpha Omega Engineering or FHC Inc.) were lowered through a 23-gauge stainless steel guide tube using a grid system (Crist Instruments). The electrodes were advanced remotely using a micromanipulator (MO-97S, Narishige) attached to the recording chamber. Signals obtained from the electrodes were amplified, bandpass filtered (300 Hz to 10 kHz), and monitored online using oscilloscopes and an audio device. Once a task-related neuron was encountered, waveforms of action potentials were isolated using software with real-time template-matching algorithms (ASD, Alpha Omega Engineering). The occurrence of each action potential was saved in files as a time stamp with the data of eye movements and visual stimuli during the experiments. We found task-related neurons in the caudal part of the cerebellar dentate nucleus (Fig. 1d). Most neurons were recorded from stereotaxic coordinates 8 mm posterior to the interaural line and 9 mm lateral to the midline in monkey I, and 8 mm posterior and 8 mm lateral in monkey J (Supplementary Fig. 1).

**Electrical microstimulation**. To examine the causal role of neuronal activity, electrical stimulation was applied to the recording sites during the synchronized saccade task. A train of 0.2 ms biphasic pulses at 333 Hz for 100 ms was delivered as electrical stimulation. The current intensity was monitored by measuring the voltage across a 1 kΩ resistor placed in series with the electrode, and was adjusted to 100 μA. Electrical stimulation was applied at 100, 200, 300, or 400 ms following every four synchronized saccades (550 ms SOA). The stimulation timing was constant during each trial (~ four times per trial), but varied from trial to trial. We have never delivered stimulation during the reactive saccade trials.

**Histological procedures**. After completion of the experiments in monkey I, several marking lesions were made by injecting direct current (10–20 μA, tip negative, ~1 mC) through electrodes placed at known coordinates. Several days after this procedure, the animal was sedated, administered analgesics, deeply anesthetized with pentobarbital, and transcardially perfused with phosphate-buffered saline followed by 3.5% paraformaldehyde. The brain was then removed, fixed, and equilibrated with 30% sucrose. Histological sections (100 μm, coronal) were cut on a freezing microtome (HM440E, Microm) and stained with cresyl violet. The recording sites were reconstructed according to stereotaxic coordinates, and the depth of the electrode tip relative to the dorsal border of the dentate nucleus, which was verified physiologically during the experiments (Fig. 1d and Supplementary Fig. 1).

**Behavioural data analysis**. The eye position signals obtained directly from the eye coil device (MEL-25, Enzanshi Kogyo) were digitized at a 16-bit resolution, sampled at 1 kHz, saved to a file during the experiments, and analysed offline using Matlab (Mathworks). During the experiments, saccades were detected online when horizontal eye position crossed the centre of the screen. However, for the offline analysis, saccade onset was detected when the angular eye velocity exceeded 200°/s and an eye displacement was >7°. We measured two temporal parameters of saccades. The inter-saccadic interval (ISI) was the onset interval of successive targeting saccades in opposite directions. The temporal error was the time from the target onset to saccade initiation and was equivalent to the latency of reactive saccades. When we examined the time course of saccade latency in each trial (Fig. 1c), the data of reactive saccades for different SOAs were plotted every 550 ms.

**Classification of neurons**. Neurons that showed periodic activity during saccades were classified according to the time course of their activity. First, we aligned the data with the sixth and subsequent synchronized saccades in each direction and calculated the spike density function (Gaussian kernel, σ = 20 ms). Next, the data at 275 ms before and after saccades in the opposite directions were concatenated and normalized by the maximum and minimum values for each neuron. Based on the time course of normalized activity, a total of 95 neurons were classified into four categories using Ward's hierarchical clustering method (Supplementary Fig. 2a). The classification was made using the data from the trials with a 550 ms SOA, and the activities in different conditions were compared across groups. The Unilateral neurons showed a preference for ipsilateral (n = 24) or contralateral (n = 9) saccades to the recording site, and data from these neurons were combined in the subsequent quantitative analysis (Fig. 2a). Unilateral, Bilateral, and Postsaccade neurons had different preferences for saccade direction, behavioural condition, and sensory stimulus, and were recorded from different depths in the cerebellar nucleus (Supplementary Fig. 1). We also performed one-way ANOVAs with three factors of saccade direction, task condition, and SOA for neuronal activity during a 200 ms period before (Unilateral and Bilateral neurons) and after (Postsaccade neuron) saccades (Supplementary Fig. 2b).

**Calculation of indices that characterize neuronal activity**. To characterize neuronal activity, we calculated the following modulation indices for individual neurons and compared them across neuronal groups. The directional index (DI) was defined as (Ipsi–Contra)/(Ipsi + Contra), where Ipsi and Contra indicate the magnitude of neuronal activity measured for ipsilateral and contralateral saccades, respectively. Neuronal activity was measured from the spike density function during a 200 ms period starting from 250 ms before (Unilateral and Bilateral neurons) or immediately after (Postsaccade neurons) saccades.

The prediction index (PI) was calculated as (Pred–React)/(Pred + React), where Pred and React represent the magnitude of firing modulation during predictive and reactive saccades in the preferred direction, respectively. For predictive saccades, the activity was measured for the sixth or later saccades in the sequence in trials with a 550 ms SOA. For reactive saccades, the data were aligned with saccades following a 550 ms fixation interval. To measure the firing modulation, we initially searched for the peak of the spike density function during ±275 ms of the saccades in the preferred direction. We then searched for the minimum value of the activity within ±275 ms of the peak. The magnitude of the firing modulation was measured as the difference between these values. Similarly, the PI for different SOAs was also calculated by measuring the firing modulation for saccades with the same length of measurement interval as the SOA. The modulation of neuronal activity for different task conditions and saccade direction were also evaluated by computing the mutual information (Supplementary Methods and Supplementary Fig. 3).

The sensorimotor index (SMI) was defined as $(Targ–Sac)/(Targ + Sac)$, where *Targ* and *Sac* denote the neuronal activity around the time of the target onset and saccade, respectively. To calculate the index, we first aligned the neuronal data with target onset or saccade initiation for the sixth and subsequent synchronized saccades in trials with a 550 ms SOA. Then the time of peak activity was determined from the spike density function computed for each alignment. Next, we measured the neuronal activity at ±50 ms from the time of the peak for each of the 2nd–5th sequences of the target (*Targ*) and saccade (*Sac*) and computed the SMI (Supplementary Fig. 10a). We mainly consider the value obtained from the second saccade because these saccades were reactive in nature and therefore the target onset and saccade initiation were temporally decoupled. If the neuronal activity was better aligned with the target onset, the SMI had a positive value. If the neuronal activity was better aligned with saccades, it had a negative value. The SMIs for the second saccades in the preferred direction were compared across neuronal groups (Fig. 5c), although one neuron was tested only in the non-preferred direction in the second sequence. The index was also computed for saccades in the opposite direction for only those Bilateral and Postsaccade neurons that exhibited bidirectional modulation (Supplementary Fig. 10b). We also performed additional analyses to complement the results obtained from the comparison of the SMIs across types of neurons. Details are described in Supplementary Methods and Supplementary Figs. 10 and 11 legends.

**Correlation between neuronal activity and saccade timing**. The cerebellum is thought to be involved in the predictive control of movement and the detection of errors that are necessary for learning. To evaluate the information carried by each dentate nuclear neuron, the correlations between neuronal activity and the timing of saccades were examined. When investigating the relationship between neuronal activity and motor control, data were aligned with synchronized saccades, and partial correlations of trial-by-trial neuronal firing rate with the timing of the next saccade (ISI) were calculated every 200 ms (10 ms steps), controlling for temporal errors of saccades and the previous ISI (Fig. 3c–e). Since the goal of this analysis was to evaluate the signals regulating saccade initiation, the partial correlation was calculated for the data until the next saccade. Therefore, the number of trials contributing to the partial correlation decreased shortly before the next target onset. To examine the relationship between neuronal activity and timing errors, data were again aligned with synchronized eye movements, and partial correlations with temporal error (saccade latency) were calculated after controlling for the previous ISI (Fig. 4c–e). The partial correlation coefficients at 200–400 ms and 0–200 ms following synchronized saccades, where large correlations were found, were used as indices for evaluating the involvement of neuronal activity in motor control and error monitoring, respectively, and the groups of neurons were compared (Fig. 3f, 4f, Supplementary Figs. 6d and 8d).

**Stimulation data analysis**. The effects of electrical microstimulation were quantitatively analysed for 49 sites where stimulation was delivered for ≥20 synchronized saccades in each condition. The effect size (Cohen's *d*) for each condition was calculated as $(\mu_{stim} − \mu_{cont})/\sqrt{[(\sigma_{stim}^2 + \sigma_{cont}^2)/2]}$, where μ and σ indicate the mean and standard deviation of the ISI with (stim) or without (cont) stimulation, respectively. The Dunnett test was used to evaluate the effect of electrical stimulation at each site and the timing of saccades in either direction. If a significant effect was found in at least one stimulation condition, the data obtained from that site were included for further analysis ($n = 32$ and 8 sites for monkeys I and J, respectively, Fig. 6b, bottom). The data from 40 stimulation sites were classified into three groups using the hierarchical clustering method based on the effect sizes under eight different stimulation conditions (Fig. 6c and d). Since the stimulus effect at each site is expected to be determined by combinations of task-related neurons in the surrounding area, we attempted to model the stimulation effect by a certain combination of the effects of stimulation to the three groups of neurons. The stimulation effects on the different groups of neurons were estimated based on simple Pearson correlations between each neuron's activity and the ISI calculated at the four electrical stimulation times and averaged for each group of neurons (Supplementary Fig. 13a–c). Because there were only nine Unilateral neurons with contralateral preference, the correlation coefficient for the population of Unilateral neurons was calculated by combining the values of the ipsilateral and contralateral neurons and flipping the same value (Supplementary Fig. 13c, green dotted lines). Since these values are considered to reflect the strength of the influence of each group's activity on saccade timing, we assumed that the effect of electrical stimulation would be a linear sum of these values with appropriate weights. Thus, the effect sizes in eight different conditions for each stimulation site (d) will be described as,

$$d = w_{ipsi}Uni_{ipsi} + w_{contra}Uni_{contra} + w_{bi}Bi + w_{post}Post \quad (1)$$

where w indicates the weight for each group of neurons and Uni, Bi, Post are the impacts of stimulation effect of each group of neurons on saccade timing calculated above as the Pearson correlation at eight different timing (Supplementary Fig. 13c). We constrained the weight (w) of each group to be positive because electrical stimulation increases the activity of nearby neurons. The values of w calculated by the least-squares method for each stimulation site were normalized and are plotted in Fig. 6e.

**Statistical analysis**. The unpaired t-test or analysis of variance (ANOVA) was used to compare the indices, peak latencies, and recording sites among the

neuronal groups, and the one sample t-test was used to compare the mean of each group with a constant value. To evaluate the effects of electrical stimulation, Dunnett's test was performed between the ISI in each stimulation condition and in the control. All tests were performed using the Statistical toolbox for Matlab. To evaluate the partial correlation between neuronal activity and saccade timing for each neuron, the distribution of the r-values obtained from the permutated data (1,000 iterations) was compared with the actual data (Fig. 3f, 4f, Supplementary Figs. 6d and 8d). Other statistical methods are described in the relevant text or figure legends.

Further details of the analysis in the Supplementary Figures can be found in the Supplementary Methods and in the relevant figure legends.

**Reporting summary**. Further information on research design is available in the Nature Research Reporting Summary linked to this article.

## Data availability

All data analysed here are included in this article and its supplementary information files. Numerical data for each figure and supplemental figures are provided in the Source Data file.

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

## Acknowledgements

The authors thank M. Suzuki for administrative help; H. Miyaguchi for animal care and training; M. Takei and M. Kusuzaki for manufacturing some equipment; and other lab members for comments, discussions, and their help on surgical and histological procedures. Animals were provided by the National Bio-Resource Project. This work was supported partly by grants from the Ministry of Education, Culture, Sports, Science and Technology of Japan (21K06418 to K.O., 15H05985 to R.T., 18H05523, 21H04810 to M.T.).

## Author contributions

K.O. analysed the data and edited and revised the manuscript. R.T. designed and performed the experiments and analysed the data. M.T. conceptualized and supervised the project, helped designing and performing the experiments, drafted and revised the manuscript. K.O. and R.T. contributed equally to this work.

## Competing interests

The authors declare no competing interests.
