## [Peer Review File · Nature Communications]

Neural signals regulating motor synchronization in the primate deep cerebellar nucleiREVIEWER COMMENTS

Reviewer #1 (Remarks to the Author):

This study investigates the response properties of cerebellar cells during a task where monkeys performed saccades in synchrony with periodically alternating visual stimuli. Different response properties were observed across three different cell classes. Notably, the cells with bilateral responses showed activity modulations associated with the timing of the next saccade, the current temporal error between the stimulus and the response, the target onset, and the predictive instead of the reactive saccadic behavior. The authors interpreted these findings as evidence that lateral cerebellum contains multiple functional modules for the acquisition of internal rhythms, predictive motor control, and error detection during synchronized movements.

This is an interesting paper providing novel notions of the role of deep cerebellar nuclei on rhythmic timing. However, I am afraid that the analyses employed are not well suited to characterize the properties of cells and some of the claims and interpretations of the paper are not fully supported by the data.

Here are my main concerns.

- 1) The classification of cells should be performed using an explicit model, such as an ANOVA, using the discharge rate as dependent parameter and the saccadic period (pre/post), saccade direction (right/left), tempo (three levels) and task condition (predictive/reactive) as factors. Furthermore, for each of these factors, the authors could compute the Mutual Information (MI), which is based on response probability distributions in order to assess response selectivity. Hence, compared to the DI, PI, and SMI, the MI is a more robust measure of selectivity of neural responses.
- 2) The classification of neurons as bilateral, unilateral and postsaccadic was done employing the data for the 550-ms SOA condition. What about the other SOAs (400 and 700)? Previous studies in cortical regions showed sequential and temporal neural activity selective for distinct inter-stimulus intervals (Merchant et al., 2013; Crowe et al., 2014).
- 3) Did you find some evidence of ramping activity (monotonic increase of activity as a function of elapsed time) that was modulated by the SOA in the time or magnitude of peak of activity or the slope of the ramping profile (Merchant et al., 2011)?
- 4) I guess the partial correlation analysis depicted in Figure 3 was carried out on the data of SOA 550ms. Before analyzing the correlation between neural activity and the ISI for one SOA, it is important to assess the selectivity of neurons across SOAs. In fact, figure 2 shows that all the activation profiles are similar across SOAs. In addition, the negative partial correlation effect of the ISI seems to simply depend on the response alignment, if the response is aligned to the next saccade probably there will not be a significant change between short and long ISI trials, since the neural responses look very stereotypic.

5) Temporal scaling. Some neurons appear to have a ramping pattern of activity (figure 3b) and the population seems to show activity patterns stretched in time for longer inter-saccade intervals (figure 3c, short vs long). It would be nice to measure the temporal scaling capabilities of the three types of neurons (Merchant and Averbeck, 2017; Wang et al., 2018).

6) I strongly suggest that the authors use the time warping analysis (Perez et al., 2013) specifically designed to determine whether the cell activity is statistically aligned to sensory or motor events during a synchronization task. The algorithm was validated with cell simulations and different neural databases, and can be a powerful tool to dissociate cells with stimulus driven or sensory predictive responses from motor neurons (Merchant et al., 2015).

7) Decoding analysis on the cell populations for saccade direction, tempo, task condition (predictive/reactive), error correction and sensory vs motor events could nicely complement the described encoding properties of cells.

8) It important to dissociate whether the effect of electrical stimulation is related with rhythmic timing from motor preparation and execution.

9) The authors should provide a summary of the similarities and differences in the encoding properties of cells between the two monkeys.

Minor comments.

1) For the introduction, please consider that the actual hypothesis in the literature is that interval-based timing depends on the cerebellum whereas beat based timing depends on the basal ganglia and both converge at the level of the medial premotor areas (Teki et al., 2017; Breska and Ivry, 2018; Merchant et al., 2015).

2) Indicate in the legend of Figure 2 the color code for the heatmaps of the activity profiles.

3) Some neurons classified as bilateral in Figure 2a seems to no respond in the ipsi-sac condition (figure 2a, left plot, last plot's rows).

4) How is the peak timing of neurons computed? Probably just the peak of response was identified, and a response onset algorithm is probably a better option.

5) The error correction signal is postsaccadic neurons is quite interesting.

Reviewer #2 (Remarks to the Author):

Predictive movements synchronized with external rhythms are common among vocal learning animals such as human and songbirds and are often observed in our daily life. In a previous study, the authors' group challenged a controversial issue about whether vocal non-learning animals can show these synchronized rhythmic movements, and found that macaque monkeys, a vocal non-learning species, exhibit predictive synchronized rhythmic saccadic eye movements (Takeya et al., 2017). In the present study, using the same behavioral paradigm, the authors attempted to uncover the neural mechanism underlying these predictive synchronized rhythmic movements.

These synchronized movements require predictive motor control. The authors' group has shown this motor control was regulated by neural networks including the basal ganglia, thalamus, and cerebellum. In the present study, they focused on the cerebellum and recorded single-unit activity from the cerebellar dentate nucleus in macaque monkeys performing predictive synchronized saccades and reactive simple saccades. They found that a subgroup of neurons in the dentate nucleus showed a greater ramping activity for predictive synchronized saccades than for reactive saccades. This activity was evoked at the onset of sensory saccadic target rather than the onset of saccadic eye movements, suggesting that the activity did not represent simple motor command. The distribution of these neurons was different from that of neurons with different activity patterns in the dentate nucleus. They also electrically stimulated recording sites in the dentate nucleus and found that the electrical stimulation affected the latency of predictive synchronized saccades. Notably, the effect of the electrical stimulation was explained by activating different subgroups of neurons. For example, stimulating at the ventral part of the dentate nucleus, where neurons showed the ramping activity for predictive synchronized saccades, facilitated both ipsilateral and contralateral saccades. These results suggest that the cerebellar dentate nucleus is involved in the acquisition of predictive movements synchronized with external rhythms.

I feel this study that can be important for the literature. The experiments are well designed. The findings are novel and presented straightforwardly and clearly. The story and conclusion are logical. However, I have some concerns that need to be addressed before final decision.

(1) They found that a subgroup of neurons in the dentate nucleus showed a greater ramping activity for predictive synchronized saccades than for reactive simple saccades. This result suggests that these dentate neurons are involved in the regulation of predictive synchronized saccades. However, another interpretation may be possible. The latency of predictive synchronized saccades was much shorter than that of reactive saccades. Thus, the greater ramping activity observed for predictive synchronized saccades might reflect a simple "facilitation effect" on saccades. This possibility can be tested by analyzing the correlation between the activity and saccade latency for both predictive synchronized saccades and reactive saccades. Especially, if there is a significant correlation between the activity and

the latency of reactive saccades, the ramping activity can be explained by the facilitative effect on saccades.

(2) The effect of electrical stimulation on predictive synchronized saccades suggests a causal contribution of the ramping activity of dentate neurons to these saccades. However, I feel that the way of presentation of data is not easy to understand. For example, the authors found that neurons in the ventral part of the dentate nucleus exhibited the greater ramping activity for predictive synchronized saccades, and mentioned that the effect of electrical stimulation on the ventral part of this nucleus facilitated both ipsilateral and contralateral saccades. The authors used a tricky correlation analysis to derive this conclusion. Instead of the correlation analysis, why don't you separate the stimulation sites into dorsal and ventral ones, and averaged the stimulation effect separately for these dorsal and ventral sites? If the authors' conclusion is correct, the averaged effect of ventral stimulation sites should exhibit a facilitation of both ipsilateral and contralateral saccades (i.e., decrease in the latency of these saccades).

(3) If the authors electrically stimulate the dentate nucleus during the reactive saccade task, are reactive saccades affected by the stimulation? According to the idea that the dentate nucleus, especially its ventral part, is involved in the acquisition of predictive synchronized movements, stimulating this location is thought to be ineffective for reactive saccades. It is hard to add new experiments in systems neuroscience research using macaque monkeys, and I don't require the new experiment. But I want to see how the authors respond to this comment.

(4) The authors examined the correlation between neuronal activity and the latency of "next" saccades, and examined the effect of electrical stimulation of the latency of "next" saccades. But I think movements synchronized with external rhythms are the "sequence" of motor actions. Therefore, there may be a correlation between neuronal activity and the latency of saccades "after the next" or even "later" saccades. The electrical stimulation may affect the latency of saccades "after the next" or even "later" saccades. Such data could strengthen the authors' conclusion.

Reviewer #3 (Remarks to the Author):

In this study, Okada and colleagues performed single-neuron recordings from the cerebellar dentate nucleus in monkeys, while the monkeys were performing synchronized saccades to periodically alternating visual stimuli or reactive saccades. Three groups of neurons were identified based on the activity before or after saccades and the preference of the saccade directions. Bilateral and Unilateral neurons are active before saccades, but with a different preference of saccade directions, and

Postsaccade neurons are active after saccades. Moreover, Bilateral neurons displayed a greater activity for synchronized than reactive saccades. Using electrical stimulation as a manipulation approach, they found stimulation site-dependent effects on predictive saccades, possibly arising from activating different groups of neurons with different strengths. The authors proposed multiple parallel functional modules in the cerebellum.

While I think the results of this paper are interesting, I do have a few major concerns and I am afraid that I cannot support this manuscript for publication without reading the next version with improved writing and seeing apparent conceptual advances.

First, the writing of this manuscript is not clear enough, making it very challenging to read and understand. The authors should re-organize their writing very carefully before the next submission so that they can convey their results and ideas to the audience. The current writing makes it hard to identify the conceptual advances.

In the introduction, the paragraphs seem to go back and forth without moving forward and deeper when introducing the background knowledge.

The results section needs a stronger storytelling flavor – why the authors did this, what the hypothesis was, what data they obtained, and what the interpretation is. To me, some paragraphs read like figure captions, and I find it difficult to link different pieces of results together because the interpretation of the results is limited in the results and discussion sections. For example, on Page 11, “Figure 5c plots the cumulative frequency distribution of the SMI for each neuron group.”

In addition, in the figure caption, the authors should add more details and convey the message a bit clearer, so that the readers don't need to go back to the main text to find out what the figure panels mean and whether the results are statistically significant. For example, Figure 6c. The message of the caption is not clear. “The box-whisker plot in the upper panel indicates the median, quartiles, and range of effect size for different stimulation times and saccade direction.” Is the effect size of the saccade direction, or the temporal error? This is confusing and the readers have to go to the main text back and forth.

Second, the conceptual advance, as well as a coherent picture, is not clear to me. How do these three groups of neurons function together, and how are they connected in the proposed model? A related point is the “internal model”. The discussion does not make clear how these findings fit within the internal model framework. Is it fit with a forward model, or inverse model, or both? And how the authors compare their results with what has been shown for entrainment tasks.

Third, electrical stimulation needs more data and analysis in order to support the hypothesis. What are the effects of electrical stimulation on reactive saccades? Will stimulation induce a delay in response timing during reactive saccades? The authors proposed multiple parallel functional modules in the cerebellum, based on electrical stimulation. But how do they define and distinguish modules? A simple separation of relative dorsal-ventral locations (yet heavily overlapping) is not enough to define modules.

Finally, the conclusion related to Fig. 5 is not convincing enough, and Supplementary Fig. 5a is confusing. The current results are not enough to support that sensory prediction and motor preparation are dissociable. To sufficiently prove this dissociation, extra experiments and analysis are needed. For example, periodic onset and offset of visual cues without the need for left and right saccades, similar to the experiment the lead author did before in (Kameda 2019 eLife). Also, it would be very helpful to see how the SMI changes as a function of the order number of target onset in the three groups of neurons, and as a function of saccade latency.

Here are minor suggestions.

1. There are a few terms that need to be better defined or explained to reach a wide neuroscientist audience. For example, the authors should explain the term “stimulus onset asynchrony (SOA)” briefly in the main text and in Figure 1 so that the general neuroscience audience will understand.

2. In Figure 6, there is an illustration of inter saccade interval (ISI), but ISI first appears in the main text related to Figure 3. I suggest moving the illustration of ISI earlier, for example, in Figure 1.

3. The authors should make it more clear how the partial correlations are calculated. Page 8: “the partial correlation between the activity of individual neurons and the inter-saccadic interval (ISI), controlling for the previous ISI and saccade latency.” I was confused because the neuronal activity is a time series while saccade latency and ISI are single numbers. Then I assume (realized) that the partial correlations were calculated using trial-by-trial data points, instead of using time series. The authors should make it clear to reduce the confusion. One way to improve the clarity is to add another panel either in the main or supplementary figures to show how the partial correlation at a given time window for a given neuron is calculated.

4. In Supplementary Fig. 1a, regarding the dorsal-ventral location of the different types of neurons, even the statistical test shows the distribution, the 3D scatter plot is not convincing. From the 3D scatter plot, it looks like the Ipsilateral neurons and Bilateral neurons tended to be in the middle or span across the entire recording depth, instead of ventrally. Perhaps the authors can use a better angle for visualization.

5. Supplementary Fig. 1 was called in the main text after Supplementary Fig. 2a, 2b, 2c.

6. Supplementary Fig. 5a, 5b were called after Supplementary Fig. 5c.

7. Supplementary Fig. 4a, 4b, were not called in the text.

8. The range of the x-axis in Fig. 3c,d are inconsistent with Fig. 3a,b. Same comments on Fig. 4a-d.

9. Three durations of SOA were used for the predictive saccades, 400, 550, and 700 ms. But in Fig. 3 & 4, only two groups are visualized, short versus long. Why not present three groups?

10. Fig. 4a, the caption “The Data are aligned with ipsilateral predictive (synchronized) saccades, and the trials are sorted by saccade latency (temporal error)” is confusing. I assume the authors mean that “The Data are from synchronized saccade experiments and trials are aligned with saccades and sorted by saccade latency.” Please clarify.

11. Fig. 5a,b, right panels: some curves are in red, and this should be mentioned in the figure caption.

12. Fig. 6f and Supplementary Fig. 6e. The authors suggested that the location of stimulation sites and their effect also follow the dorsal-ventral trend, similar to the relative location of Bilateral and Postsaccade neurons. To make this statement more convincing, the authors should directly compare the locations between the stimulation sites and recorded neurons.

13. “Saccade latency” and “temporal errors” in the text were used interchangeably. The authors can make the definition clearly first and then stick to one of them so that it will cause less confusion to the readers.

14. “Synchronized saccades” and “predictive saccades” were used interchangeably. The authors should be consistent and stick to one of them to reduce confusion.

15. Figure 6c. How do the raw data (e.g. individual data points) look like? It would be great if the authors could show the data used to calculate the Cohen’s d before this panel c.

16. Page 13: “We then attempted to account for the changes in saccade timing due to electrical stimulation by linearly adding these data together (Methods).” And the related Page 25: “where w indicates the weight of signals and Uni, Bi, Post are the signals carried by each group of neurons calculated above (Supplementary Fig. 6c).” I read multiple times but still don’t understand how these (Uni, Bi, Post) are calculated. Please clarify.

17. Statements related to Supplementary Fig. 7 are not convincing. The relative dorsal-ventral locations of neurons and their anatomical connections across different studies cannot be simply combined to draw the conclusion without further verification.

Responses to Reviewer #1's comments:

This study investigates the response properties of cerebellar cells during a task where monkeys performed saccades in synchrony with periodically alternating visual stimuli. Different response properties were observed across three different cell classes. Notably, the cells with bilateral responses showed activity modulations associated with the timing of the next saccade, the current temporal error between the stimulus and the response, the target onset, and the predictive instead of the reactive saccadic behavior. The authors interpreted these findings as evidence that lateral cerebellum contains multiple functional modules for the acquisition of internal rhythms, predictive motor control, and error detection during synchronized movements.

This is an interesting paper providing novel notions of the role of deep cerebellar nuclei on rhythmic timing. However, I am afraid that the analyses employed are not well suited to characterize the properties of cells and some of the claims and interpretations of the paper are not fully supported by the data.

Response: We thank this reviewer for his/her careful reading and insightful comments on our manuscript. We performed many additional analyses as suggested by this reviewer. We believe that the revised manuscript has been greatly improved.

Here are my main concerns.

1) *The classification of cells should be performed using an explicit model, such as ANOVA, using the discharge rate as dependent parameter and the saccadic period (pre/post), saccade direction (right/left), tempo (three levels) and task condition (predictive/reactive) as factors. Furthermore, for each of these factors, the authors could compute the Mutual Information (MI), which is based on response probability distributions in order to assess response selectivity. Hence, compared to the DI, PI, and SMI, the MI is a more robust measure of selectivity of neural responses.*

Response: In response to this comment, we performed ANOVAs and also computed the mutual information for individual neurons. These results are summarized in Supplementary Figs. 2b (ANOVA) and 3d–e (mutual information), respectively. Accordingly, we have modified the relevant text (pages 6–7). As shown in Supplementary Fig. 2b, many Bilateral neurons showed a significant directionality because ANOVA considers statistical difference in firing rate between conditions but does not reflect their relative magnitude. The cluster analysis considers the similarity of the time course of neuronal activity around saccades, rather than only the statistical difference of neuronal activity at specific time intervals. Therefore, we think that cluster analysis is more appropriate to characterize the properties of each neuron than ANOVA, and we would like to maintain this classification method.

2) *The classification of neurons as bilateral, unilateral and postsaccadic was done employing the data for the 550-ms SOA condition. What about the other SOAs (400 and 700)? Previous studies in cortical regions showed sequential and temporal neural activity selective for distinct inter-stimulus intervals (Merchant et al., 2013; Crowe et al., 2014).*

Response: We calculated the two indices (DI and PI) for the data of 400 and 700-ms SOAs

(Supplementary Fig. 5a and b) to see if each group of neurons exhibited similar response properties across SOAs. The results show that the distributions of these indices were comparable for different SOAs. These results are reported in the relevant text and Supplementary Fig. 5 legend. We also thoroughly examined the possible interval selectivity by comparing the time course of preparatory activity of Bilateral and Unilateral neurons (Supplementary Fig. 4; please see the responses to the Major comments #3 and #5 below). Briefly, only a few neurons showed a preference for the 550-ms condition (i.e., tuned representation), while most neurons showed temporal scaling of ramping activity for different SOAs.

3) *Did you find some evidence of ramping activity (monotonic increase of activity as a function of elapsed time) that was modulated by the SOA in the time or magnitude of peak of activity or the slope of the ramping profile (Merchant et al., 2011)?*

Response: For each of Unilateral and Bilateral neurons, we examined whether the time and magnitude of peak activity and the slope of ramping activity differed across SOAs. Supplementary Fig. 4d reports the number of neurons with a significant difference based on one-way ANOVAs. As stated in the figure legend, the post-hoc multiple comparisons showed that only a few neurons showed distinct activity for 550-ms SOA only ("For almost all neurons, these changes were scaled orderly except for a few neurons exhibiting a selective change only for the 550-ms SOA (one Unilateral neuron for peak timing, one Unilateral and two Bilateral neurons for peak magnitude).").

Supplementary Figs. 4a and b compare the time courses of population activity for different SOAs. In the population, Bilateral neurons changed the ramp slope only, while Unilateral neurons changed both the ramp slope and peak timing for different SOAs. These results are reported in the revised text (page 7). When temporally scaling the preparatory activity, the time course of activity for different SOAs appears to be very similar for both Bilateral and Unilateral neurons (Supplementary Fig. 4c).

4) *I guess the partial correlation analysis depicted in Figure 3 was carried out on the data of SOA 550m. Before analyzing the correlation between neural activity and the ISI for one SOA, it is important to assess the selectivity of neurons across SOAs. In fact, figure 2 show that all the activation profiles are similar across SOAs. In addition, the negative partial correlation effect of the ISI seems to simply depend on the response alignment, if the response is aligned to the next saccade probably there will not be a significant change between short and long ISI trials, since the neural responses look very stereotypic.*

Response: As stated in the responses to the Major comments #2 and #3 above, we thoroughly examined the tempo selectivity by comparing the indices (DI and PI) and ramping activity across SOAs. In the revised manuscript, we also plot the time courses of the partial correlation computed for trials with 400 and 700-ms SOAs (Supplementary Fig. 5c and d), which are consistent with those calculated for 550-ms SOA. In addition, we computed the partial correlation with the ISI for the data aligned with the next saccade (Supplementary Fig. 7b). We found significant partial correlation with the ISI for either alignment, although the correlation disappeared around the time of saccade (± 100 ms, gray shading) because the neuronal activity reached the peak irrespective of saccade timing. We

have added some note in the Supplementary Fig. 7 legend ("Since the peak of the preparatory activity aligned with the next saccades was similar between trials with early and late saccades, the partial correlation coefficients around the time of saccade (± 100 ms, gray shading) were not different from zero ($P > 0.05$).").

5) *Temporal scaling. Some neurons appear to have a ramping pattern of activity (figure 3b) and the population seems to show activity patterns stretched in time for longer inter-saccade intervals (figure 3c, short vs long). It would be nice to measure the temporal scaling capabilities of the three types of neurons (Merchant and Averbeck, 2017; Wang et al., 2018).*

Response: As explained in the response to the Major comment #3 above, the preparatory activity of both Bilateral and Unilateral neurons appeared to be temporally scaled. This can be clearly seen in Supplementary Fig. 4c. In response to this comment, we have added some lines to note the similarity to the previous studies (page 7, "The temporal scaling of preparatory activity was consistent with the previous studies in the medial frontal cortex and striatum, as well as those in the cerebellum in the range of hundreds of milliseconds.").

6) *I strongly suggest that the authors use the time warping analysis (Perez et al., 2013) specifically designed to determine whether the cell activity is statistically aligned to sensory or motor events during a synchronization task. The algorithm was validated with cell simulations and different neural databases, and can be a powerful tool to dissociate cells with stimulus driven or sensory predictive responses from motor neurons (Merchant et al., 2015).*

Response: The time warping analysis is useful to dissociate sensory from motor response. However, because synchronized movement is triggered by sensory *prediction* rather than by actual sensory signals, the sensory prediction and motor response are not dissociable. Movement signals may be separated from sensory prediction signals only during the transition from reactive to synchronized movements, where sensory prediction must precede movement.

We applied the time warping analysis to both during synchronized saccades (the 6th and subsequent saccades in sequence) and during the transition (the 2nd saccade) to see if the sensory and motor-aligned activities altered. Supplementary Fig. 11a shows that for all types of neurons the time-warping indices (log likelihood ratio of sensory vs. motor alignment) calculated for synchronized saccades had a negative value, indicating that the neuronal activity was better aligned with motor execution or sensory prediction, rather than actual target timing. Furthermore, for Bilateral neurons only, the time-warping indices during the transition differed from those during synchronization (paired *t*-test, $P < 0.003$), suggesting that Bilateral neurons may signal sensory prediction rather than movement per se. The time-warping index during the transition well correlated with the SMI (Supplementary Fig. 11a, bottom panel, $r = 0.54$). Thus, during the transition from reactive to synchronized saccades, the activity of Bilateral neurons predicted the target timing, no matter whether the magnitude of activity (SMI) or the trial-by-trial variability of spike occurrence (time warping analysis) was considered. We have inserted some lines to report this fact (page 12).

7) *Decoding analysis on the cell populations for saccade direction, tempo, task condition*

(predictive/reactive), error correction and sensory vs motor events could nicely complement the described encoding properties of cells.

Response: As suggested, we performed the decoding analysis on the population of each type of neurons (Supplementary Fig. 9). The data show that saccade direction and tempo (SOA) can be well decoded from either population, while the task condition and temporal error were better decoded from Bilateral neurons than the other types of neurons. These results were consistent with a significant correlation of Bilateral neurons with saccade parameters, supporting that these neurons play an important role in synchronized saccades.

8) It is important to dissociate whether the effect of electrical stimulation is related with rhythmic timing from motor preparation and execution.

Response: Because electrical stimulation at many sites altered saccade timing bilaterally, these effects were different from perturbation of specific motor command or directional bias of motor preparation. In relation to this, and in response to the other reviewers, we considered possible causes of electrical stimulation during reactive saccades by calculating the correlation between neuronal activity and the latency of reactive saccades. As stated in the last paragraph of the revised Results, our data indicate that Bilateral neurons regulate synchronized saccades in both directions while they seem to regulate only contralateral reactive saccades. These results suggest that Bilateral neurons may represent internal rhythms, thereby regulating the timing of synchronized saccades in both directions.

9) The authors should provide a summary of the similarities and differences in the encoding properties of cells between the two monkeys.

Response: We now report the number of each type of neurons recorded from each animal and compared the DI and PI between the animals. These results are now reported in the legends of Supplementary Fig. 2 ("Red, green, and blue clusters indicate Bilateral ($n = 33$, $n = 22$ and 11 for monkeys I and J, respectively), Unilateral ($n = 33$, $n = 28$ and 5 for I and J, respectively), and Postsaccade neurons ($n = 29$, $n = 22$ and 7 for I and J, respectively), respectively.") and Supplementary Fig. 3 ("A two-way ANOVA (neuron types \times monkeys) for each index showed no significant difference between monkeys (DI, $F_{1,94} = 0.04$, $P = 0.84$; PI, $F_{1,90} = 0.1$, $P = 0.76$); peak timing, $F_{1,93} = 0.8$, $P = 0.37$).").

Minor comments.

1) For the introduction, please consider that the actual hypothesis in the literature is that interval-based timing depends on the cerebellum whereas beat based timing depends on the basal ganglia and both converge at the level of the medial premotor areas (Teki et al., 2017; Breska and Ivry, 2018; Merchant et al., 2015).

Response: We have modified a sentence to state that the previous studies suggest a role for the basal ganglia in beat-based timing (page 3, "Evidence from clinical cases, functional imaging with non-motor rhythmic tasks, and physiological experiments in primates all suggest a role for the basal ganglia in beat-based timing.").

2) *Indicate in the legend of Figure 2 the color code for the heatmaps of the activity profiles.*

Response: We have added this information. "Yellow and blue indicate higher and lower activity, respectively (*parula* colormap, MATLAB)."

3) *Some neurons classified as bilateral in Figure 2a seems to no respond in the ipsi-sac condition (figure 2a, left plot, last plot's rows).*

Response: As mentioned in the response to the Major comment #1 above, the cluster analysis considers the time course of neuronal activity rather than the statistical difference in neuronal activity at specific task periods. The Bilateral neurons seemed to be classified based not only on the activity before saccades in both directions but also on the time course of activity after saccades (compare the activity of Bilateral neurons after ipsilateral saccades with that of Unilateral neurons after saccades in the non-preferred direction). Nevertheless, the amount of directionality quantified by the DI and mutual information differed between these groups of neurons (Supplementary Fig. 3a and d).

4) *How is the peak timing of neurons computed? Probably just the peak of response was identified, and a response onset algorithm is probably a better option.*

Response: We simply detected the peak of the spike density profile aligned with saccade initiation. We have added this information to the legend of Supplementary Fig. 3b ("Timing of peak activity was measured from the spike density profile aligned with synchronized saccades in the preferred direction."). Since many neurons exhibited ramping activity, the peak timing is more reliable measure than the onset timing.

5) *The error correction signal is postsaccadic neurons is quite interesting.*

Response: Thank you for this supportive comment.

Responses to Reviewer #2's comments:

Predictive movements synchronized with external rhythms are common among vocal learning animals such as human and songbirds and are often observed in our daily life. In a previous study, the authors' group challenged a controversial issue about whether vocal non-learning animals can show these synchronize rhythmic movements, and found that macaque monkeys, a vocal non-learning species, exhibit predictive synchronized rhythmic saccadic eye movements (Takeya et al., 2017). In the present study, using the same behavioral paradigm, the authors attempted to uncover the neural mechanism underlying these predictive synchronized rhythmic movements.

These synchronized movements require predictive motor control. The authors' group has shown this motor control was regulated by neural networks including the basal ganglia, thalamus, and cerebellum. In the present study, they focused on the cerebellum and recorded single-unit activity from the cerebellar dentate nucleus in macaque monkeys performing predictive synchronized saccades and reactive simple saccades. They found that a subgroup of neurons in the dentate nucleus showed a greater ramping activity for predictive synchronized saccades than for reactive saccades. This activity was evoked at the onset of sensory saccadic target rather than the onset of saccadic eye movements, suggesting that the activity did not represent simple motor command. The distribution of these neurons was different from that of neurons with different activity patterns in the dentate nucleus. They also electrically stimulated recording sites in the dentate nucleus and found that the electrical stimulation affected the latency of predictive synchronized saccades. Notably, the effect of the electrical stimulation was explained by activating different subgroups of neurons. For example, stimulating at the ventral part of the dentate nucleus, where neurons showed the ramping activity for predictive synchronized saccades, facilitated both ipsilateral and contralateral saccades. These results suggest that the cerebellar dentate nucleus is involved in the acquisition of predictive movements synchronized with external rhythms.

I feel this study that can be important for the literature. The experiments are well designed. The findings are novel and presented straightforwardly and clearly. The story and conclusion are logical. However, I have some concerns that need to be addressed before final decision.

Response: We thank the reviewer for his/her positive comments. All requested information has been added.

(1) They found that a subgroup of neurons in the dentate nucleus showed a greater ramping activity for predictive synchronized saccades than for reactive simple saccades. This result suggests that these dentate neurons are involved in the regulation of predictive synchronized saccades. However, another interpretation may be possible. The latency of predictive synchronized saccades was much shorter than that of reactive saccades. Thus, the greater ramping activity observed for predictive synchronized saccades might reflect a simple "facilitation effect" on saccades. This possibility can be tested by analyzing the correlation between the activity and saccade latency for both predictive synchronized saccades and reactive saccades. Especially, if there is a significant correlation between the activity and the latency of reactive saccades, the ramping activity can be explained by the facilitative effect on saccades.

Response: As seen in the population activity in Fig. 3c and d, the preparatory activity during synchronized saccades started just after the preceding saccades in both Unilateral and Bilateral neurons. This can be more clearly seen in newly added Supplementary Fig. 4. Therefore, it is unlikely that synchronized saccades required hastened, larger facilitation than reactive saccades.

In response to this and the other reviewer's comments, we computed the correlation between neuronal activity and the latency of reactive saccades (Supplementary Fig. 14a). We found that Postsaccade and Unilateral neurons showed significant correlation in a similar way to that during synchronized saccades (Supplementary Fig. 13b), suggesting that they may regulate the timing of both types of saccades. However, Bilateral neurons showed significant correlation for contralateral reactive saccades only, while they showed correlation for synchronized saccades in both directions. These results indicate that the signals conveyed by Bilateral neurons may not be strictly related to movement like other types of neurons.

(2) The effect of electrical stimulation on predictive synchronized saccades suggests a causal contribution of the ramping activity of dentate neurons to these saccades. However, I feel that the way of presentation of data is not easy to understand. For example, the authors found that neurons in the ventral part of the dentate nucleus exhibited the greater ramping activity for predictive synchronized saccades, and mentioned that the effect of electrical stimulation on the ventral part of this nucleus facilitated both ipsilateral and contralateral saccades. The authors used a tricky correlation analysis to derive this conclusion. Instead of the correlation analysis, why don't you separate the stimulation sites into dorsal and ventral ones, and averaged the stimulation effect separately for these dorsal and ventral sites? If the authors' conclusion is correct, the averaged effect of ventral stimulation sites should exhibit a facilitation of both ipsilateral and contralateral saccades (i.e., decrease in the latency of these saccades).

Response: The reviewer is right. We performed the direct comparison of stimulation effects between the dorsal and ventral sites. Supplementary Fig. 13f shows that electrical stimulation to the dorsal sites delayed ipsilateral saccades, whereas that to the ventral sites facilitated saccades in both directions. These results are consistent with the fact that neurons with different properties were found at different depths in the nucleus. We have added this information to the revised text (page 14, bottom line).

(3) If the authors electrically stimulate the dentate nucleus during the reactive saccade task, are reactive saccades affected by the stimulation? According to the idea that the dentate nucleus, especially its ventral part, is involved in the acquisition of predictive synchronized movements, stimulating this location is thought to be ineffective for reactive saccades. It is hard to add new experiments in systems neuroscience research using macaque monkeys, and I don't require the new experiment. But I want to see how the authors respond to this comment.

Response: As quantified by the prediction index (PI), the magnitude of activities during synchronized and reactive saccades were comparable in Unilateral and Postsaccade neurons. Furthermore, we found that the activity of these neurons before target presentation in the reactive task correlated with saccade latency in a similar way to those in the

synchronized task (Supplementary Fig. 13a). These results suggest that electrical stimulation of these neurons may change the latency of reactive saccades. The correlation analysis also showed that the activity of Bilateral neurons inversely correlated with the latency of contralateral reactive saccades but not with the latency of ipsilateral saccades (Supplementary Fig. 13a). These results suggest that electrical stimulation to the ventral part of the dentate nucleus (where Bilateral neurons reside) may promote only contralateral reactive saccades, while the same stimulation promoted synchronized saccades in both directions (Fig. 6d and Supplementary Fig. 13f). We have added this information to the last paragraph in the Results (page 15).

(4) The authors examined the correlation between neuronal activity and the latency of “next” saccades, and examined the effect of electrical stimulation of the latency of “next” saccades. But I think movements synchronized with external rhythms are the “sequence” of motor actions. Therefore, there may be a correlation between neuronal activity and the latency of saccades “after the next” or even “later” saccades. The electrical stimulation may affect the latency of saccades “after the next” or even “later” saccades. Such data could strengthen the authors’ conclusion.

Response: In response to this important comment, we examined the effect of electrical stimulation on the timing of the second saccade after stimulation (Supplementary Fig. 12a). As expected, the stimulation changed saccade timing. However, this may be due to the inverse correlation of successive saccade timing during synchronization, as can be seen from the mirror image of the direction of changes compared to Fig. 6b. In fact, the effect size of electrical stimulation calculated for successive saccades inversely correlated (Supplementary Fig. 12b). To assess the direct effect of electrical stimulation on the timing of two later saccades, we attempted to predict the intersaccadic interval (ISI) in trials with electrical stimulation from the relationship of successive ISIs in trials without stimulation in the same session (Supplementary Fig. 12c). Supplementary Fig. 12d summarizes the normalized prediction error of saccade timing before and after electrical stimulation, and that for the next saccades. The results showed that the timing of the second saccade after stimulation was mostly explained by the inverse correlation of successive ISIs and was indistinguishable from that before stimulation, indicating that electrical stimulation exerted only an immediate effect.

In addition, to test whether neuronal activity correlated with the timing of two later saccades, the time course of the partial correlation shown in Fig. 3e was extended backward in time to the previous ISI (Supplementary Fig. 7c). The data show that, when the correlation between successive saccade timing was controlled, neuronal activity in the cerebellar nucleus seems to regulate only the next saccade timing. We have added this information to the revised manuscript and modified the text accordingly (pages 9 and 14).

Responses to Reviewer #3's comments:

In this study, Okada and colleagues performed single-neuron recordings from the cerebellar dentate nucleus in monkeys, while the monkeys were performing synchronized saccades to periodically alternating visual stimuli or reactive saccades. Three groups of neurons were identified based on the activity before or after saccades and the preference of the saccade directions. Bilateral and Unilateral neurons are active before saccades, but with a different preference of saccade directions, and Postsaccade neurons are active after saccades. Moreover, Bilateral neurons displayed a greater activity for synchronized than reactive saccades. Using electrical stimulation as a manipulation approach, they found stimulation site-dependent effects on predictive saccades, possibly arising from activating different groups of neurons with different strengths. The authors proposed multiple parallel functional modules in the cerebellum.

While I think the results of this paper are interesting, I do have a few major concerns and I am afraid that I cannot support this manuscript for publication without reading the next version with improved writing and seeing apparent conceptual advances.

Response: We appreciate this reviewer's many insightful comments on our manuscript. We have tried to make clearer the points raised by the reviewer.

First, the writing of this manuscript is not clear enough, making it very challenging to read and understand. The authors should re-organize their writing very carefully before the next submission so that they can convey their results and ideas to the audience. The current writing makes it hard to identify the conceptual advances.

In the introduction, the paragraphs seem to go back and forth without moving forward and deeper when introducing the background knowledge.

The results section needs a stronger storytelling flavor – why the authors did this, what the hypothesis was, what data they obtained, and what the interpretation is. To me, some paragraphs read like figure captions, and I find it difficult to link different pieces of results together because the interpretation of the results is limited in the results and discussion sections. For example, on Page 11, “Figure 5c plots the cumulative frequency distribution of the SMI for each neuron group.”

In addition, in the figure caption, the authors should add more details and convey the message a bit clearer, so that the readers don't need to go back to the main text to find out what the figure panels mean and whether the results are statistically significant. For example, Figure 6c. The message of the caption is not clear. “The box-whisker plot in the upper panel indicates the median, quartiles, and range of effect size for different stimulation times and saccade direction.” Is the effect size of the saccade direction, or the temporal error? This is confusing and the readers have to go to the main text back and forth.

Response: Considering these comments, we have modified many parts of the text and figure captions to make the points clearer. The portions of actual changes in the text are highlighted with red fonts.

Second, the conceptual advance, as well as a coherent picture, is not clear to me. How do these three groups of neurons function together, and how are they connected in the proposed model? A related point is the "internal model". The discussion does not make clear how these findings fit within the internal model framework. Is it fit with a forward model, or inverse model, or both? And how the authors compare their results with what has been shown for entrainment tasks.

Response: Although we had to shorten the text because of word limit, we believe that the revised Discussion directly explains the conceptual advances and provides a coherent picture. Along with the neuronal signals related to motor control and error monitoring, we found in this study that neurons in the cerebellar dentate nucleus also represent internal rhythm of periodic visual stimuli. The results are consistent with our previous finding of entrained activity in the same region of the cerebellar nuclei.

Third, electrical stimulation needs more data and analysis in order to support the hypothesis. What are the effects of electrical stimulation on reactive saccades? Will stimulation induce a delay in response timing during reactive saccades? The authors proposed multiple parallel functional modules in the cerebellum, based on electrical stimulation. But how do they define and distinguish modules? A simple separation of relative dorsal-ventral locations (yet heavily overlapping) is not enough to define modules.

Response: We did not apply electrical stimulation during reactive saccades, as explicitly stated in the revised text. Because the magnitude of activity for Unilateral and Postsaccade neurons were similar between reactive and synchronized saccades (as indexed by the Prediction Index), these neurons may play a role in both saccades.

To understand how electrical stimulation will change the timing of reactive saccade, we calculated a simple correlation between neuronal activity and the timing of the next reactive saccade. Supplementary Fig. 14a shows that the activity of Unilateral and Postsaccade neurons before target presentation in the reactive task correlated with saccade latency, indicating that electrical stimulation of these neurons may change the latency of reactive saccades. The data also show that the activity of Bilateral neurons inversely correlated with the latency of contralateral reactive saccades but not with that of ipsilateral saccades. These results suggest that electrical stimulation to the ventral part of the dentate nucleus (where Bilateral neurons reside) promote only contralateral reactive saccades, while the same stimulation promoted synchronized saccades in both directions (Fig. 6d and Supplementary Fig. 13f). We have added a paragraph to the end of the Results to explain these possibilities (page 15).

In response to the comments of this and the other reviewer, we now plot the effects of electrical stimulation separately according to the depth of the stimulation sites (Supplementary Fig. 13f). Although the distributions of different types of neurons heavily overlap (as pointed out by this reviewer), the effects of electrical stimulation were distinct between the dorsal and ventral portions of the dentate nucleus. Along with the different anatomical projections between the dorsal and ventral dentate nucleus (Strick et al., 2009), our results of different properties of neuronal activity and stimulation effect strongly support our hypothesis that there are multiple (rather than single) functional modules in the

nucleus for the control of synchronized saccades.

Finally, the conclusion related to Fig. 5 is not convincing enough, and Supplementary Fig. 5a is confusing. The current results are not enough to support that sensory prediction and motor preparation are dissociable. To sufficiently prove this dissociation, extra experiments and analysis are needed. For example, periodic onset and offset of visual cues without the need for left and right saccades, similar to the experiment the lead author did before in (Kameda 2019 eLife). Also, it would be very helpful to see how the SMI changes as a function of the order number of target onset in the three groups of neurons, and as a function of saccade latency.

Response: During synchronized movements, sensory prediction and motor execution are difficult to dissociate because the former triggers the latter. However, during the transition from reactive to predictive saccades at the beginning of each trial, sensory prediction must precede movement and these two may be separated. In response to this comment, we have made the following changes to the manuscript.

First, we have strengthened the description of the analysis in Fig. 5 and the previous Supplementary Fig. 5a (now Supplementary Fig. 11b). Although neurons recorded in this study were not examined in the task performed in the previous experiments, we have added a note to the Discussion that neurons in the posterior portion of the dentate nucleus exhibit predictive activity that peaks at the timing of periodic stimuli (page 16).

Second, as suggested by this reviewer, we computed the SMI for the 2nd through the 5th saccades in the sequence (Supplementary Fig. 10a). These data were also separated into four groups according to temporal error (saccade latency). When the target order was early and the temporal error was long (reactive saccades), the activity of Bilateral neurons tended to be better aligned with the target than saccade, resulting in larger SMI values.

Third, to quantify the sensory and motor alignments of neuronal activity in a different way, we compared the time course rather than the magnitude of activity between early reactive (2nd saccade) and later synchronized saccades (> 5th, Supplementary Fig. 10c). For Bilateral neurons, the rank correlation coefficient calculated between the spike density profiles aligned with the target onset tended to be large, while that calculated for spike densities aligned with saccades tended to be small. For Unilateral neurons, this relationship reversed. These results further support our findings that Bilateral neurons carry sensory prediction signals while Unilateral neurons carry movement signals.

Finally, as suggested by the reviewer #1 (major comment #6), we also performed the time-warping analysis (Perez, Kass, Merchant, 2013). Briefly, in this analysis, the times of target onset or saccade were aligned across trials by scaling the time axis, and the means of probability density of spike timing was computed. Then, the maximum likelihood of spike timing in each trial was calculated to quantify whether the neuronal activity was better aligned with the sensory or motor events (Supplementary Fig. 11a). Bilateral neurons predominantly exhibited motor rather than sensory alignment during synchronized saccades (because movement occurred at the time of sensory prediction rather than actual target onset), but shifted the response toward the sensory alignment during the early reactive saccades (2nd saccades in the sequence). By contrast, the activity of Unilateral neurons was better aligned with movement regardless of the task condition. These results further support that the activity of Bilateral neurons was better aligned with the sensory prediction rather than actual motor execution. The text has been revised accordingly to

incorporate this information (page 12).

Here are minor suggestions.

1. There are a few terms that need to be better defined or explained to reach a wide neuroscientist audience. For example, the authors should explain the term “stimulus onset asynchrony (SOA)” briefly in the main text and in Figure 1 so that the general neuroscience audience will understand.

Response: We have added some words in the text and Figure 1 legend to explain the SOA.

2. In Figure 6, there is an illustration of inter saccade interval (ISI), but ISI first appears in the main text related to Figure 3. I suggest moving the illustration of ISI earlier, for example, in Figure 1.

Response: We have made this change. The definitions of ISI and temporal error are now illustrated in Figure 1b.

3. The authors should make it more clear how the partial correlations are calculated. Page 8: “the partial correlation between the activity of individual neurons and the inter-saccadic interval (ISI), controlling for the previous ISI and saccade latency.” I was confused because the neuronal activity is a time series while saccade latency and ISI are single numbers. Then I assume (realized) that the partial correlations were calculated using trial-by-trial data points, instead of using time series. The authors should make it clear to reduce the confusion. One way to improve the clarity is to add another panel either in the main or supplementary figures to show how the partial correlation at a given time window for a given neuron is calculated.

Response: We have inserted the term "trial-by-trial" to the text relevant to Figures 3 and 4. In addition, we have added panels to Supplementary Figs. 6a and 8a to explain the procedures of calculating the partial correlation.

4. In Supplementary Fig. 1a, regarding the dorsal-ventral location of the different types of neurons, even the statistical test shows the distribution, the 3D scatter plot is not convincing. From the 3D scatter plot, it looks like the Ipsilateral neurons and Bilateral neurons tended to be in the middle or span across the entire recording depth, instead of ventrally. Perhaps the authors can use a better angle for visualization.

Response: In response to this comment, we separately plot the data of different types of neurons (Supplementary Fig. 1a). The different distributions are now clear.

5. Supplementary Fig. 1 was called in the main text after Supplementary Fig. 2a, 2b, 2c.

6. Supplementary Fig. 5a, 5b were called after Supplementary Fig. 5c.

7. Supplementary Fig. 4a, 4b, were not called in the text.

Response: We have corrected these problems by revising the text and changing the order of Supplementary figures and panels.

8. *The range of the x-axis in Fig. 3c,d are inconsistent with Fig. 3a,b. Same comments on Fig. 4a-d.*

Response: We apologize for the lack of important information in the original manuscript. To remove the effects of activity after saccades, the partial correlation was computed for the data before the next saccades. We have added this information to the Methods (page 25, "Since the goal of this analysis was to evaluate the signals regulating saccade initiation, the partial correlation was calculated for the data until the next saccade. Therefore, the number of trials contributing to the partial correlation decreased shortly before the next target onset."). Because of this reason, we do not show the partial correlation shortly before the next saccades (approximately 550 ms in most trials). This does not affect the quantitative analyses in Figs. 3f and 4f because the time window for the analysis was up to 400 ms from the saccade.

9. *Three durations of SOA were used for the predictive saccades, 400, 550, and 700 ms. But in Fig. 3 & 4, only two groups are visualized, short versus long. Why not present three groups?*

Response: Figures 3 and 4 plot the data for 550-ms SOA only, and the time courses of neuronal activity are shown for one-third of trials with the longer and shorter intersaccadic intervals. The data for different SOAs are now shown in Supplementary Fig. 5. We have added this information to the revised text and Figs. 3 and 4 legends.

10. *Fig. 4a, the caption "The Data are aligned with ipsilateral predictive (synchronized) saccades, and the trials are sorted by saccade latency (temporal error)" is confusing. I assume the authors mean that "The Data are from synchronized saccade experiments and trials are aligned with saccades and sorted by saccade latency." Please clarify.*

Response: We modified the sentence to make the point clearer ("The data are from synchronized saccade experiments (SOA 550 ms) and trials are aligned with ipsilateral saccades and sorted by temporal error (saccade latency)."). We also modified the sentence in Fig. 3a caption accordingly.

11. *Fig. 5a,b, right panels: some curves are in red, and this should be mentioned in the figure caption.*

Response: We have added a sentence to Figure 5a caption ("Red curves indicate the responses to the second target or saccade in the preferred direction, which were used for the quantitative analysis in c.").

12. *Fig. 6f and Supplementary Fig. 6e. The authors suggested that the location of stimulation sites and their effect also follow the dorsal-ventral trend, similar to the relative location of Bilateral and Postsaccade neurons. To make this statement more convincing, the authors should directly compare the locations between the stimulation sites and recorded neurons.*

Response: Since the locations of recorded neurons and the stimulation sites along the dorsal-ventral axis are shown in Supplementary Figs. 1b and 13e, respectively, we would

like to avoid redundant presentation where these panels are replotted in a new figure. As stated in the response to the Major comment #3 above, Supplementary Fig. 13f now shows that the stimulation effects at the dorsal and ventral sites were qualitatively different.

13. *“Saccade latency” and “temporal errors” in the text were used interchangeably. The authors can make the definition clearly first and then stick to one of them so that it will cause less confusion to the readers.*

14. *“Synchronized saccades” and “predictive saccades” were used interchangeably. The authors should be consistent and stick to one of them to reduce confusion.*

Response: We have made these changes. We now mainly use the terms "temporal error" and "synchronized saccades" and use parentheses when "latency" or "prediction" is needed for clarity.

15. *Figure 6c. How do the raw data (e.g. individual data points) look like? It would be great if the authors could show the data used to calculate the Cohen's d before this panel c.*

Response: In response to this comment, we have added histograms of saccade timing for the data in Figure 6a (previous Fig. 6b), along with the effect size for each condition.

16. *Page 13: “We then attempted to account for the changes in saccade timing due to electrical stimulation by linearly adding these data together (Methods).” And the related Page 25: “where w indicates the weight of signals and Uni , Bi , $Post$ are the signals carried by each group of neurons calculated above (Supplementary Fig. 6c).” I read multiple times but still don't understand how these (Uni , Bi , $Post$) are calculated. Please clarify.*

Response: We have modified sentences in the Results and the Methods to make the point clearer.

17. *Statements related to Supplementary Fig. 7 are not convincing. The relative dorsal-ventral locations of neurons and their anatomical connections across different studies cannot be simply combined to draw the conclusion without further verification.*

Response: As stated in the title of this Supplementary figure (now Supplementary Fig. 15), the diagram is not the conclusion but is a hypothesis drawn from our observations and previous anatomical studies. We would like to retain this diagram which will provide a running hypothesis for future studies. We have added some words to the text accordingly.

REVIEWERS' COMMENTS

Reviewer #1 (Remarks to the Author):

The authors did an excellent job at incorporating the reviewers' comments on the new version of the paper. Different metrics were used to validate the original indexes and now the claims of the manuscript are better supported by the data. I just have a set on minor comments:

- 1) The Mutual information for saccade direction should have a range from 0 to $\log_2(2)$ which is 1.
- 2) Please state which classification method was used for decoding.
- 3) Is not immediately clear why the three types of cells showed a very large decoding accuracy for saccade direction across a very wide time window, please discuss in the supplementary section.

Reviewer #2 (Remarks to the Author):

The authors adequately responded to all my concerns in the revised manuscript. I have only a minor comment to their response as follows.

In the previous review, I pointed out the possibility that the greater ramping activity observed for predictive synchronized saccades compared with reactive saccades might reflect a simple "facilitation effect" on saccades, because the latency of the predictive synchronized saccades was shorter than that of the reactive saccades. To exclude this possibility, I required them to calculate the correlation between the ramping activity and the latency of the reactive saccades. The authors showed this correlation analysis in Supplementary Fig. 14 of the revised manuscript, and found that some neuron types did not show the correlation for the reactive saccades. Based on this data, the authors concluded that "These results indicate that the signals conveyed by Bilateral neurons may not be strictly related to movement like other types of neurons", although this description was not mentioned in the revised manuscript. I feel that it is very important for this study to discriminate whether the ramping activity of dentate neurons is involved in the generation of synchronized saccades or in the general facilitation of saccades. I therefore recommend the authors to include their consideration in the manuscript.

Reviewer #3 (Remarks to the Author):

The authors have done an excellent job in revising their manuscript. They addressed all my major and minor concerns. I appreciate their effort in adding motivation, clarity, and interpretation of their results in more detail. They also re-organized the figures according to the main text and added extra supplementary figures to better support the story. I found the new version of the manuscript more much accessible and convincing. I support the publication of this manuscript in Nature Communications.

Responses to Reviewer #1's comments:

1) *The Mutual information for saccade direction should have a range from 0 to $\log_2(2)$ which is 1.*

Response: As noted in the Supplementary Information Methods, the sign of MI was assigned to compare with the DI and PI. We have modified the sentence in SI Methods, added a word to the axis labels in SI Fig. 3d and e, and inserted sentence to the figure legend for clarification (SI Methods, "For the comparison with the DI and PI, the sign of the MI was assigned so that the preference for ipsilateral saccades or synchronized task was positive and the preference for contralateral saccades or reactive task was negative."; SI Fig. 3 legend, "The mutual information is signed so that the preference for contralateral saccades is negative." and "The mutual information is signed so that the preference for the reactive saccade task is negative.").

2) *Please state which classification method was used for decoding.*

Response: We used a maximum correlation coefficient classifier for the decoding analysis in SI Fig. 9. We have added some words to SI Methods and the relevant figure legend to clarify this point (SI Methods, "For each decoding run, the data was divided into five sets of trials, and the maximum correlation coefficient classifier was trained on the four sets and tested for the remaining.", "... using the same classification method (maximum correlation coefficient classifier)").

3) *Is not immediately clear why the three types of cells showed a very large decoding accuracy for saccade direction across a very wide time window, please discuss in the supplementary section.*

Response: We have added a sentence to the legend of Supplementary Fig. 9e ("Saccade direction could be decoded reliably from the population activity of all types of neurons, consistent with the ANOVA results shown in Supplementary Fig. 2b").

Responses to Reviewer #2's comments:

In the previous review, I pointed out the possibility that the greater ramping activity observed for predictive synchronized saccades compared with reactive saccades might reflect a simple "facilitation effect" on saccades, because the latency of the predictive synchronized saccades was shorter than that of the reactive saccades. To exclude this possibility, I required them to calculate the correlation between the ramping activity and the latency of the reactive saccades. The authors showed this correlation analysis in Supplementary Fig. 14 of the revised manuscript, and found that some neuron types did not show the correlation for the reactive saccades. Based on this data, the authors concluded that "These results indicate that the signals conveyed by Bilateral neurons may not be strictly related to movement like other types of neurons", although this description was not mentioned in the revised manuscript. I feel that it is very important for this study to discriminate whether the ramping activity of dentate neurons

is involved in the generation of synchronized saccades or in the general facilitation of saccades. I therefore recommend the authors to include their consideration in the manuscript.

Response: As suggested, we have clarified the point in the revised text (page 15, "The correlation analysis also showed that the activity of Bilateral neurons inversely correlated with the latency of contralateral reactive saccades but not with the latency of ipsilateral saccades, whereas the same neurons were significantly correlated with the timing of synchronized saccades in both directions (Supplementary Fig. 13b). These results indicate that the signals conveyed by Bilateral neurons may not be strictly related to movement like the other types of neurons.").

Responses to Reviewer #3's comments:

The authors have done an excellent job in revising their manuscript. They addressed all my major and minor concerns. I appreciate their effort in adding motivation, clarity, and interpretation of their results in more detail. They also re-organized the figures according to the main text and added extra supplementary figures to better support the story. I found the new version of the manuscript more much accessible and convincing. I support the publication of this manuscript in Nature Communications.

Response: We thank this reviewer for his/her supportive comments.